# Uncovering hidden protein modifications with native top-down mass spectrometry

Jack L. Bennett[1,2], Tarick J. El-Baba [1,2], Konstantin C. Zouboulis[1,2], Carla Kirschbaum [1,2], Haigang Song[1,2], Frances I. Butroid[1,2], Justin L. P. Benesch [1,2], Corinne A. Lutomski[1,2] ✉ & Carol V. Robinson [1,2] ✉

Protein modifications drive dynamic cellular processes by modulating biomolecular interactions, yet capturing these modifications within their native structural context remains a significant challenge. Native top-down mass spectrometry promises to preserve the critical link between modifications and interactions. However, current methods often fail to detect uncharacterized or low-abundance modifications, limiting insights into proteoform diversity. To address this gap, we introduce precise and accurate Identification Of Native proteoforms (precisION), an interactive end-to-end software package that leverages a robust, data-driven fragment-level open search to detect, localize and quantify 'hidden' modifications within intact protein complexes. Applying precisION to four therapeutically relevant targets—PDE6, ACE2, osteopontin (SPP1) and a GABA transporter (GAT1)—we discover undocumented phosphorylation, glycosylation and lipidation, and resolve previously uninterpretable density in an electron cryo-microscopy map of GAT1. As an open-source software package, precisION offers an intuitive means for interpreting complex protein fragmentation data. This tool will empower the community to unlock the potential of native top-down mass spectrometry, advancing integrative structural biology, molecular pathology and drug development.

Post-translational modifications (PTMs) regulate biomolecular function by modulating protein dynamics, interactions and localization patterns[1,2]. Distinctly modified protein isoforms (proteoforms) act as unique biological effectors, enabling dynamic control of cellular phenotypes[3,4]. Mass spectrometry (MS)-based proteomics can correlate specific proteoforms with cellular states by identifying and quantifying modified proteins[5–7]. However, because proteins are denatured or proteolyzed, it remains challenging to directly connect such modifications with their impacts on protein structure, interactions and function.

Native top-down mass spectrometry (nTDMS) is emerging as a powerful approach for the molecular characterization of intact proteoforms and their complexes[8–11]. In a single experiment, native MS can observe and quantify protein assemblies, then fragment these complexes to define the precise molecular composition of individual subunits[8,12–17] (Fig. 1a). nTDMS promises to directly link changes in protein sequence or modification state to their effects on interactions, such as glycan-mediated stabilization of the heterodimeric human follicle-stimulating hormone[18] or modulation of protein–drug interactions through lipidation of retinal G proteins and phosphodiesterases[19].

Recent advances in instrumentation[13,16,20–22] have propelled nTDMS toward the study of proteoform-specific interactions in complex cellular and tissue environments[9,19,23–25]. However, these intricate systems present unique challenges that complicate accurate, informative and reproducible interpretation of mass spectral data, hindering discovery of new biology and definitive characterization of proteoforms[26]. In particular, despite ongoing efforts to streamline sample preparation, and ensure high-quality spectra[27–30], many biomolecular complexes are inherently heterogeneous, preventing unambiguous determination of

[1]Department of Chemistry, Physical and Theoretical Chemistry Laboratory, University of Oxford, Oxford, UK. [2]Kavli Institute for Nanoscience Discovery, University of Oxford, Oxford, UK. ✉e-mail: corinne.lutomski@chem.ox.ac.uk; carol.robinson@chem.ox.ac.uk

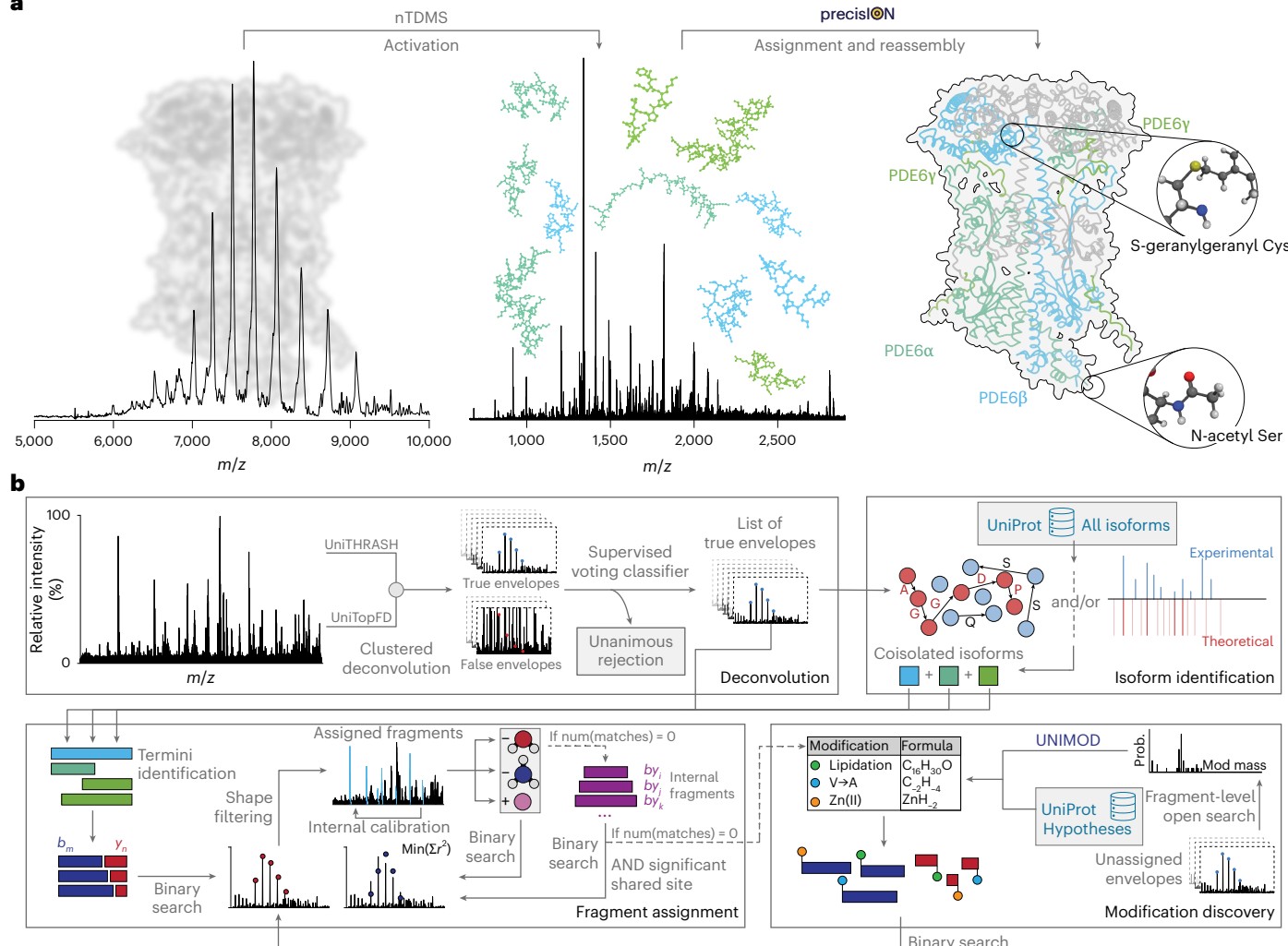

**Fig. 1 | Data-driven interpretation of native top-down mass spectra in precisION. a**, Overview of an nTDMS measurement, illustrated here with endogenous bovine PDE6, a heterotetrameric complex composed of three distinct subunits. Intact protein complexes, exhibiting peak widths of 10–1,000-Da full width at half maximum in deconvolved spectra, are selected and activated in the gas phase, yielding fragment ions measured with both: (1) sufficient resolving power to distinguish between similar protein modifications, and (2) high mass accuracy (errors typically less than 20 mDa) to confidently identify these modifications. precisION interprets the resulting spectra, assigning (modified) sequence ions which can be reassembled to uncover the diversity of proteoforms assembled within the antecedent complexes. Here, all three subunits of bovine PDE6 were identified from isolated bovine rod outer segment membranes after the intact protein complex was activated using infrared photons. Co- and post-translational modifications including

N-terminal methionine exclusion, N-terminal acetylation and cysteine lipidation were confidently identified from the fragment mass spectrum alone. Regions corresponding to the observed fragments are highlighted on the AlphaFold 3 (ref. 74) structure of the assembled complex. **b**, Schematic overview of the workflows employed by precisION for spectral deconvolution, protein isoform identification, fragment assignment and modification discovery. Isotopic envelopes are first picked and filtered (deconvolution), before the identity of the fragmented protein complex is determined using de novo sequencing and/or an open database search (isoform identification). Unmodified terminal fragments are then assigned through a semi-supervised hierarchical scheme (fragment assignment) before a fragment-level open search is used to discover and assign sets of internal fragments and co-/post-translationally modified terminal fragments (modification discovery). Mod mass, modification mass; Prob., probability.

individual subunit masses and consequent detection of protein modifications at the intact protein level[31,32]. While reducing heterogeneity through (bio)chemical methods (for example, delipidation, deglycosylation or in-solution disassembly) is appealing, such approaches may disrupt complexes, and ultimately contradict the goal of complete molecular characterization. Instead, embracing the complexity of these systems is essential, and the challenge lies in extracting meaningful information from these inherently complex nTDMS spectra.

To extend the limits of nTDMS analyses, we developed algorithmic methods to discover protein modifications from native top-down mass spectra. These approaches are integrated in precise and accurate Identification Of Native proteoforms (precisION), an interactive open-source software package that implements a robust informatic

framework designed to analyze complex proteoform assemblies (Fig. 1). Through a hierarchical approach to spectral analysis that yields a comprehensive and parsimonious interpretation of the data, precisION enables objective, semi-automated analyses of fragments from intact proteins. Building upon this platform, precisION can discover hidden protein modifications within the unannotated dark matter[33] of native top-down mass spectra, expanding analyses beyond established biological knowledge reported in databases (Fig. 1b).

Herein, we apply precisION to uncover the molecular complexity of several mammalian proteins subject to extensive co- and post-translational modification. Specifically, we characterize proteoforms of a heavily glycosylated cell surface protein, truncated signaling factor and transmembrane solute carrier. We discover

diverse PTMs that are hidden at the intact protein level due to inherent limitations in resolution of native mass spectra. precisION enables extensive assignment of (modified) terminal and internal fragments, while maintaining an ion-level false discovery rate (FDR) of less than 5%. Moreover, precisION's fragment-level open search localizes protein truncations and modifications that would not be detected using established approaches, extending the capabilities of nTDMS in the structural elucidation of highly complex biomolecular systems such as mammalian membrane proteins.

## Results

### Data-driven insight into proteoform structure and reactivity

We reasoned that complete characterization of nTDMS spectra would enable individual protein modifications, truncations and variants to be confidently identified, localized and quantified using protein fragments alone (Fig. 1a and Extended Data Fig. 1). To achieve this goal, we developed an informatic framework, precisION, optimized to address unique challenges faced in nTDMS analysis, including: (1) difficulties in obtaining accurate intact masses of proteoform subunits due to sample heterogeneity, extensive adduction or complex formation (many complexes cannot be completely disassembled in the gas phase); (2) reduced protein fragmentation compared with denaturing analyses, due to low charging and inefficient fragment dissociation; and (3) lower signal-to-noise ratios relative to denaturing measurements, due to reduced transmission efficiencies and signal dilution (for example, from extensive neutral losses or adduction) (Extended Data Fig. 2). Despite these challenges, the insights obtained by maintaining the full complement of modifications and interactions—such as subunit stoichiometry, ligand binding and responses to modifications (for example, for the endogenous PDE6 protein complex; Extended Data Fig. 1 and Supplementary Note 1)—underscore the need for a comprehensive analysis platform that extends beyond current practices.

**Overview of precisION.** precisION is structured around four primary modules (Fig. 1b), each designed to perform specialized operations, namely: (1) deconvolution of low signal-to-noise spectra, paired with envelope classification to distinguish real fragment ions from artifacts; (2) initial protein isoform identification through de novo sequencing and/or open database searches; (3) first-pass assignment of unmodified protein fragments via a hierarchical assignment scheme; and (4) second-pass assignment of modified protein fragments without requiring previous knowledge of the protein's intact mass, composition or potential modifications. For a comprehensive description of precisION see Methods and below.

High-resolution native top-down mass spectra are first deconvolved using a modified Richardson–Lucy algorithm (Fig. 1b and Supplementary Fig. 1). Then, clusters of peaks (that is, isotopic envelopes) corresponding to protein fragments are identified with the widely used algorithms TopFD[34] and THRASH[35,36]. The outputs of these algorithms are compiled into a comprehensive list of protein fragments. Due to the presence of noise and interference in raw spectra, these lists often include artifactual isotopic envelopes that cannot be reliably distinguished from the surrounding signal. We observed that scoring measures used by most established algorithms often struggle to differentiate real and artifactual envelopes when fixed thresholds are applied (Extended Data Fig. 3 and Supplementary Fig. 2). precisION therefore filters lists of fragments using a machine learning-based supervised voting classifier trained on spectrum-specific putative isotopic envelopes. Unlike existing machine learning algorithms for envelope filtering[37,38], precisION's classifier is specifically tuned to minimize the number of true envelopes that are incorrectly discarded.

To identify protein complex/es, precisION searches the refined data against databases of mature protein isoforms and their unprocessed precursors (Fig. 1b). Our observations[16], along with those of others[39,40], suggest that fragmentation in nTDMS can be driven either

by structure, where residues along a structural feature of the protein are cleaved, or by sequence, where selective cleavage occurs between residues with a higher propensity for fragmentation. To account for both mechanisms, precisION offers two approaches: it can apply a graph-based de novo sequencing algorithm to identify sequence tags in cases of structure-driven fragmentation[41], or it can perform an open search with unlimited precursor tolerance (that is, no intact mass) in cases of sequence-driven fragmentation[42–44]. Putative open search matches are ranked using a scoring metric tailored for native proteoform–spectrum matches, enhancing search sensitivity[39].

Following identification of the *m/z*-selected protein complex/es, precisION assigns ions through a hierarchical scheme based on our empirical understanding of native protein fragmentation and the biology of proteoform maturation (Fig. 1b). Initially, the algorithm assigns ions that are most likely to be observed and removes them from the fragment list before assigning those with lower probabilities. Throughout this annotation process, assigned ions serve as internal calibrants, minimizing mass errors and allowing for tight mass tolerances in successive assignment steps (Supplementary Fig. 3). Furthermore, to ensure robust fragment assignment, poorly fitting fragment matches are visually inspected by the user before assignment. This approach ensures exhaustive and accurate assignment of native top-down mass spectra, given an ample understanding of ion chemistry and the underlying protein biology.

**precisION's fragment-level open search.** When developing precisION, we initially relied on proteoform and PTM databases to identify modified sequence ions. This strategy is employed in most standard approaches. However, in many cases, a large proportion of fragments remained unassigned (Supplementary Fig. 4). We postulated that the unassigned fragments were modified sequence ions, formed through uncharacterized biological processes and/or unusual gas-phase reactivity.

To discover, assign and localize such modifications, precisION implements a fragment-level open search to identify sets of sequence ions that share a common modification (Fig. 1b). A variable mass offset is applied to each protein terminus and the number of matching fragments is evaluated (Extended Data Fig. 4). Offsets resulting in a statistically significant number of matches (expectation values calculated from a Poisson distribution; Methods) are then assigned to specific modifications using UniMod, UniProt and/or elemental composition calculators (Extended Data Fig. 4). By recognizing these patterns of modification, precisION can unveil novel proteoforms.

The fragment-level open search described above can be used to detect relatively small PTMs, as well as larger sequence ion modifications, such as protein truncation or secondary gas-phase fragmentation (that is, internal fragments). For relatively small modifications (<2,000 Da), it is feasible to conduct a quasi-continuous scan of the mass offset domain and identify all significant offsets within the tested range (Extended Data Fig. 4). However, for larger modifications, scanning continuously across a very broad range of mass offsets (>5,000 Da) is time-intensive. Thus, to discover protein truncations and sets of internal fragments that share a terminal fragmentation site (which often correspond to mass shifts >10 kDa), precisION uses a multinotch fragment-level open search[45]. Notches are selected to correspond to masses consistent with successive cleavage of amino acid residues from the N or C terminus of the protein (Supplementary Fig. 5). Together, the continuous scan and multinotch search enable detection, assignment and localization of solution- and gas-phase modifications ranging from less than 1 Da to over 10 kDa.

### Mapping glycosylation onto the human ACE2 receptor

We first sought to use precisION to comprehensively interpret fragmentation spectra for a glycoprotein not amenable to standard top-down MS analyses. Glycans are mediators of many protein–protein

interactions and proteoform-resolved analysis of glycoproteins can reveal glycan-specific effects on assembly (for example, dimerization[32,46]). However, during nTDMS, there is a possibility of simultaneous cleavage of glycans and the protein backbone. These confounding fragmentation events increase the number of theoretical fragments, making it challenging to predict and confidently identify resulting sequence ions. Here, we selected an isoform of human angiotensin-converting enzyme 2 (ACE2; Fig. 2a) since it contains six putative N-linked glycosylation sites per subunit and the occupancy of glycosylation sites influences its dimerization[46]. Specifically, distinct glycoforms of ACE2 have been shown to be differentially incorporated into dimers[46]. We modulated N-glycan occupancy through site-directed mutagenesis and expressed soluble ectodomains of various ACE2 glycoforms. Even when produced in HEK293 GNTI$^{-/-}$ cells, in which all N-glycans have a fixed composition (Man$_5$GlcNAc$_2$), complex native MS peak patterns were observed for ACE2 dimers, suggesting heterogeneity (Fig. 2b).

To uncover the basis for such heterogeneity, we examined the ACE2 glycoforms using nTDMS and applied precisION. Two major charge state distributions were observed in the MS$^1$ spectrum, corresponding to monomeric and dimeric forms of the enzyme (Fig. 2b). Focusing on the dimer, multiple sets of peaks separated by 1,210–1,220 Da were assigned as distinct ACE2 proteoforms that differ in N-glycan occupancy. The two major glycoforms were assigned as containing 10–11 glycans, consistent with high occupancy at five of the six N-glycan sites on ACE2, and partial occupancy at Asn690 (ref. 46). However, the measured mass was ~820 Da less than expected after accounting for glycans. In addition, finer peaks within each glycoform (140–160 Da with even spacing) implied an additional level of heterogeneity that could not be identified using the MS$^1$ spectrum alone.

To identify these unknown proteoforms of ACE2, we selected ACE2 dimers with 10 or 11 glycans (27+, m/z 6,565–6,685) and activated them using stepped collision energy higher-energy collisional dissociation (sceHCD). The resulting spectra (Fig. 2b) contained a plethora of fragment ions that we assigned as unmodified b-type ions using precisION. Surprisingly, we could not detect any ions containing the protein's expected C terminus. Using the fragment-level open search, we found that this lack of coverage arose from variable proteolytic processing of the C terminus of ACE2 (Fig. 2c). Subsequently, quantifying the abundance of truncated species using fragment data, we observed close agreement with the abundances estimated from the MS$^1$ spectrum (Fig. 2c, inset). This supports use of fragments for the quantification of small PTMs and their consequent proteoforms.

After assigning the complete set of unmodified b- and y-type ions, many fragments remained unassigned. Applying precisION's open search module, we identified statistically significant sets of internal fragments that share a terminal fragmentation site (E value < 0.01; Extended Data Fig. 5 and Fig. 2d). At higher collision energies, larger numbers of internal fragment sets were observed (Extended Data Fig. 5). The 'shared termini' filter employed in our approach reduced the internal fragment FDR from more than 40% to less than 6.7% across all ACE2 spectra (Extended Data Fig. 6). This approach enabled reliable interpretation of internal fragments without need for additional experimental validation or stricter mass tolerances (Extended Data Fig. 6 and Supplementary Fig. 6). As a result, we greatly increased the fraction of the spectrum that could be readily interpreted, particularly at high higher-energy collisional dissociation (HCD) energies. Consequently, we assigned an additional 632 fragment ions, giving a final sequence coverage of 30.9% (an increase from 10.6% when not considering internal fragments) for the dimeric 171-kDa complex (Fig. 2d and Supplementary Fig. 4).

To identify sequence ions with glycan modifications, we then conducted a fragment-level open search to identify ion modifications between −100 and 1,300 Da (Fig. 2e). The N-terminal mass offset scans contained several significant offsets which were made clear by using a

high-quality set of filtered isotopic envelopes (Supplementary Fig. 7) and a tight mass tolerance (3 ppm; Supplementary Fig. 8). Peaks corresponding to monoisotopic errors (1.0-mDa error) and carbon monoxide loss/a-type ions (0.9-mDa error) were observed, the latter restricted to sequence ions containing Asp24 (Fig. 2e). At higher masses, peaks corresponding to intact Hex$_5$HexNAc$_2$ (5.1-mDa error) and HexNAc (8.5-mDa error) were also detected; these ions were mapped to confirm occupancy at the first N-glycan sequon (Fig. 2d). These HexNAc-containing sequence ions arise from partial decomposition of intact glycans during collisional activation, as previously observed for glycopeptides and proteins[19,47]. We have also observed similar reactivity with denatured glycoproteins (Extended Data Fig. 7). A sodiated B$_4$ ion, corresponding to the complementary released glycan, was also detected, confirming this assignment (Fig. 2b).

## Capturing extensive modification of a cell signaling glycoprotein

Heterogeneous glycosylation presents substantial challenges in native MS, as it typically prevents resolution of individual proteoform complexes. While advanced deconvolution methods[32], charge-detection MS[48] and ion–ion reactions[49,50] can derive intact mass distributions, small protein modifications (for example, phosphorylation) are often masked by higher molecular weight glycans.

Secreted phosphoprotein 1 (SPP1), also known as osteopontin, is a signaling factor that is extensively modified during its trafficking to the extracellular matrix[51] (Fig. 3a). Different maturation processes, including O-glycosylation, truncation and phosphorylation, generate a diverse array of SPP1 proteoforms which may differentially modulate distinct signaling pathways[52] related to tissue renewal[53] and tumor growth[54]. The extensive and undefined proteoform array limits our knowledge of their impact on these therapeutically important processes.

To gain insight into the array of SPP1 proteoforms, we characterized the modification state of a recombinant form of SPP1 expressed in HEK293 cells. The mass spectrum displayed broad, unresolved peaks, indicative of substantial heterogeneity attributed to variable O-glycosylation (Fig. 3b). To penetrate the glycans and uncover underlying modifications, we selected a large range of proteoforms (m/z 3,400 ± 600) and activated the resulting ensemble using sceHCD. The resulting spectra were rich in fragments, enabling detailed investigation with precisION (Supplementary Figs. 9 and 10).

We first characterized the termini using the multinotch fragment-level open search described above (Fig. 3c). Alongside signal peptide cleavage, we observed two distinct C-terminal truncations at residues Glu197|Leu and Tyr246|Lys, as well as the unprocessed C terminus. Sequence ion abundances suggested the full-length protein to be the predominant form, followed by the Tyr246 and Glu197 truncation variants. To explore potential mechanisms underlying these truncations, we searched the MEROPS peptidase database[55] for proteases capable of cleaving at these sites. Neither site returned a definite match, suggesting nonspecific protease activity or intrinsic hydrolysis. Notably, cleavage at Glu197|Leu is consistent with the activity of matrix metalloproteases, some of which demonstrate preferences for leucine in the P1′ position[56]. The Tyr246|Lys cleavage, although less likely to involve matrix metalloprotease activity, has been previously observed in human urine, indicating its potential physiological relevance[51]. Neither of these cleavage sites correspond to the canonical thrombin cleavage event known to drive SPP1 binding to integrins (Fig. 3c).

Beyond truncation variants, precisION's open search also revealed a series of phosphorylated sequence ions originating from the N and C termini (Fig. 3d and Supplementary Fig. 11). Despite the 53 sites on SPP1 that can be phosphorylated, multiply phosphorylated ions were detected at very low frequencies, suggesting relatively low phosphorylation levels overall (Fig. 3b,d). To quantify these modifications, we acquired a set of eight spectra with varying isolation windows and

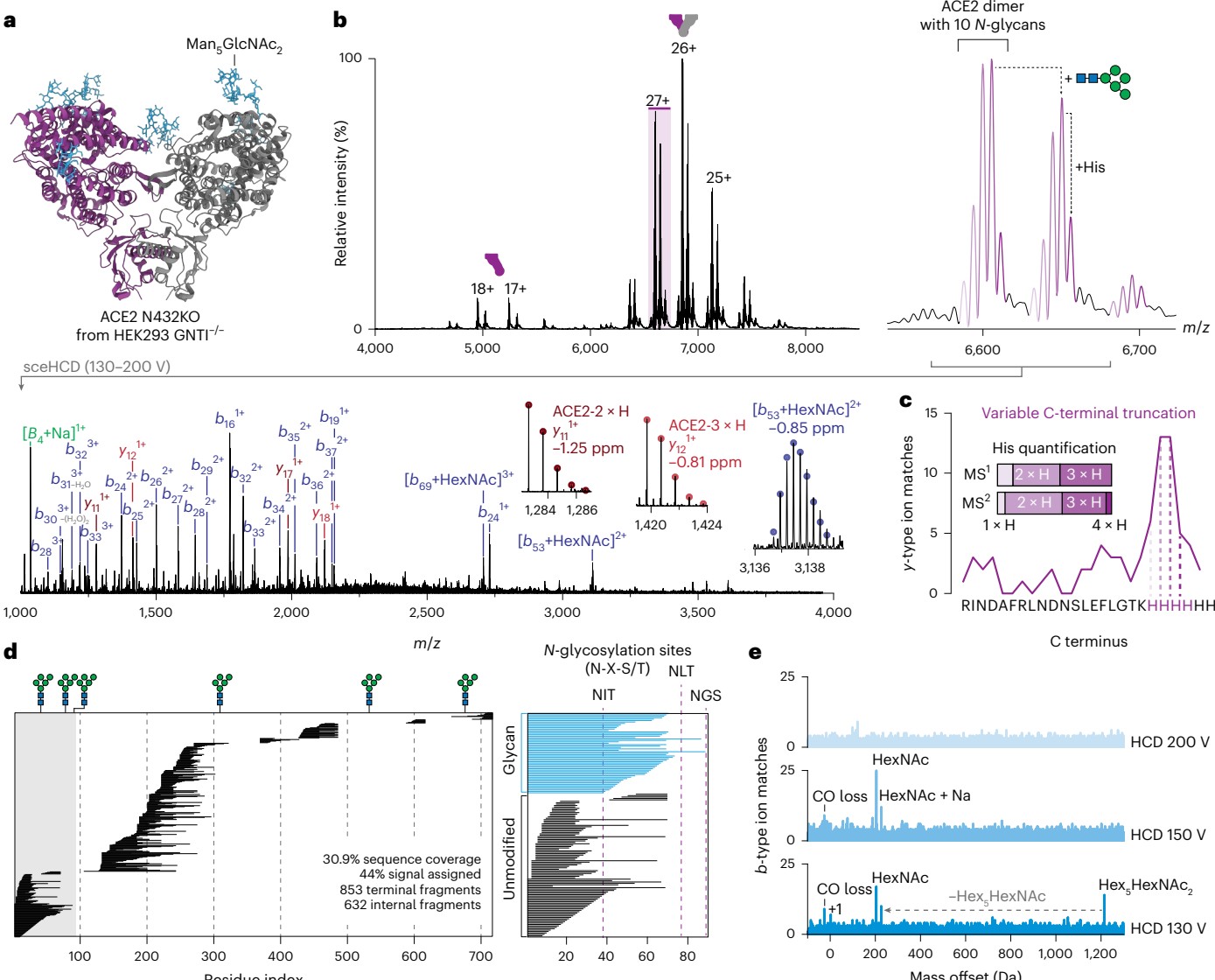

**Fig. 2 | precisION localizes *N*-glycosylation on the dimeric human ACE2.**
**a**, AlphaFold 3 structure of the ACE2 N432KO ectodomain dimer produced in HEK293 GNTI$^{-/-}$ cells. Man$_5$GlcNAc$_2$ glycans are displayed in blue. **b**, Mass spectrum of ACE2 N432KO (300 mM ammonium acetate, pH 7.0). An ensemble of ions assigned to the ACE2 dimer with 10–11 *N*-glycans and other unknown modifications (27+) were selected using the quadrupole and activated using ion–neutral collisions (HCD 130–200 V). The annotated MS$^2$ spectrum generated at an HCD acceleration voltage of 150 V is displayed below. Inset are isotopic envelopes corresponding to sequence ions arising from distinctly modified forms of ACE2. **c**, Multinotch fragment-level open search used to identify the C terminus of the ACE2 construct. Variable proteolytic processing was observed. MS$^1$- and MS$^2$-based quantifications of the different truncated forms are shown

inset. **d**, Fragment map illustrating the position of the fragmentation sites along the backbone of ACE2. Individual horizontal lines correspond to sequence ions assigned from the combined HCD dataset. *N*-glycosylation sites detected using bottom-up proteomics are displayed above the map. In some cases, we observed unmodified fragments enclosing glycosylation sites—this is likely due to the complete loss of the glycan upon activation. The right panel displays the positions of glycosylated and nonglycosylated sequence ions along the N terminus of ACE2. *N*-glycan sequons (N-X-S/T) are highlighted with dashed lines. **e**, Fragment-level open search results for the ACE2 dimer at different HCD acceleration voltages. Multiple significant offsets were observed. NGS, Asn-Gly-Ser; NIT, Asn-Ile-Thr; NLT, Asn-Leu-Thr.

relatively low HCD acceleration voltages. We then calculated the average number of phosphate groups between each residue and its nearest terminus (Fig. 3e and Supplementary Fig. 12). Our analysis indicated low phosphorylation levels across the observable regions of SPP1, with the first 116 residue region containing on average ~0.6 phosphate groups, the majority of which were localized between Asn59 and Glu80. We also observed similar results using electron transfer dissociation with supplemental collisional activation, which is known to better preserve phosphorylation during fragmentation[57] (Supplementary Fig. 13). We further validated our results enzymatically by removing *O*-glycans from the protein to reveal the average number of phosphates across the

entire length of the protein (~2.0), as well as the presence of truncated proteoforms (Extended Data Fig. 8).

### Localizing diverse lipidation on a human GABA transporter

γ-aminobutyric acid (GABA) is the primary inhibitory neurotransmitter in the adult human central nervous system[58]. At synapses, GABA binds to ionotropic[59] and metabotropic[60] receptors on the postsynaptic neuron, reducing its excitability (Fig. 4a). The sodium- and chloride-dependent GABA transporter 1 (GAT1/*SLC6A1*) plays a key role in regulating GABAergic signaling by transporting GABA back into the presynaptic neuron and surrounding glial cells following its

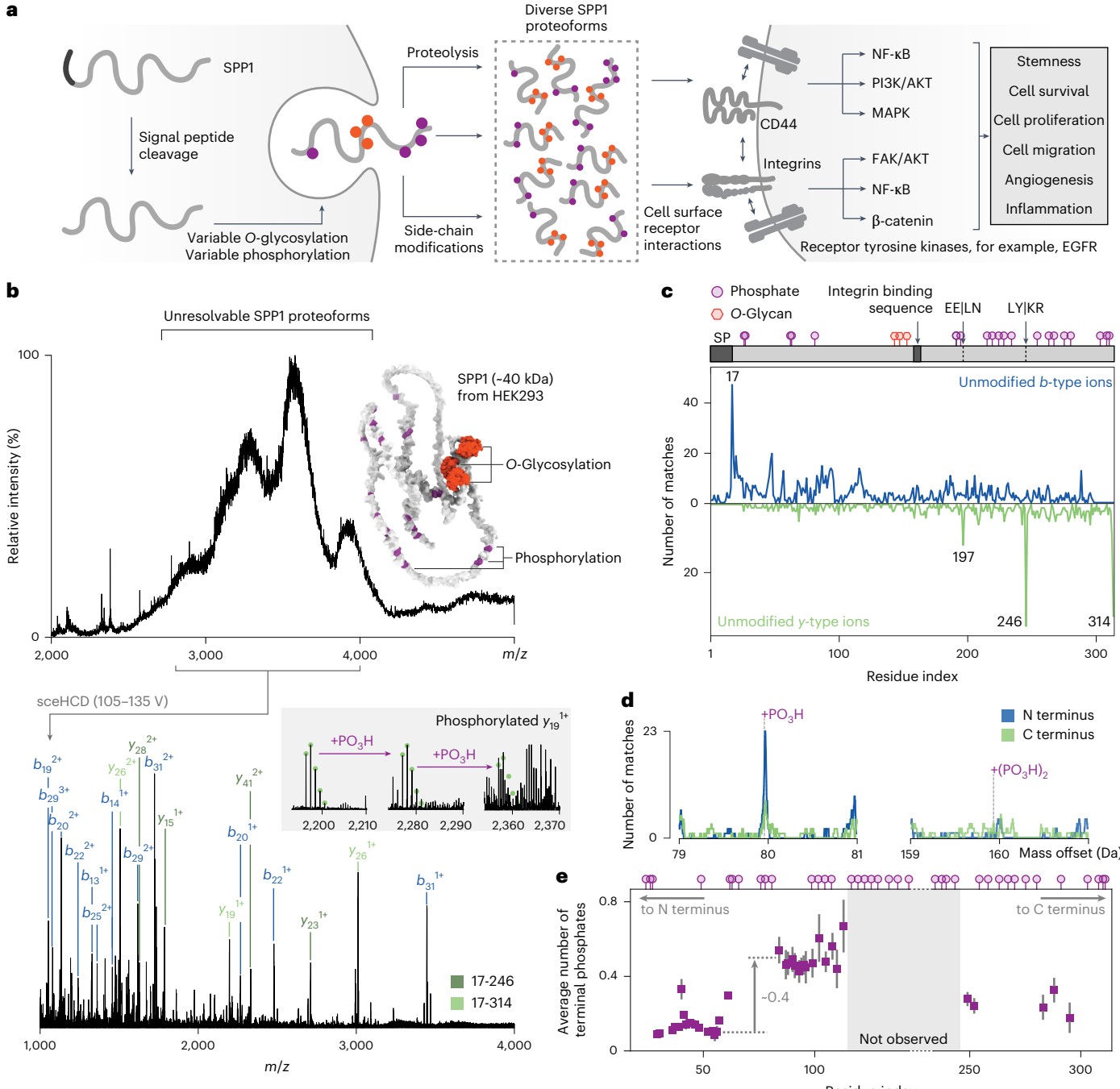

**Fig. 3 | Human SPP1 is variably truncated and phosphorylated beneath a layer of glycosylation. a**, Schematic illustrating the maturation and function of human SPP1. During and after secretion, SPP1 is extensively modified to generate a diverse range of proteoforms that interact with receptors on surrounding cells. **b**, Mass spectrum of SPP1 (1 M ammonium acetate, pH 7.0). A broad range of ions ($m/z$ 3,400 ± 600) were selected using the quadrupole and activated using ion–neutral collisions (sceHCD 105–135 V). The annotated $MS^2$ spectrum generated at an HCD acceleration voltage of 120 V is displayed below. Two truncated forms of the protein are annotated in different shades of green. Inset are the isotopic envelopes detected for $y_{19}^{1+}$ with 0, 1 or 2 phosphate groups. Intensities are scaled in each case to ensure each envelope can be clearly observed. **c**, Multinotch fragment-level open search used to identify the termini of SPP1. Data from three HCD acceleration voltages were combined before conducting the search. The upper plot displays the $b$-type ion search used to

characterize the N terminus, while the lower plot displays the $y$-type ion search used to characterize the C terminus. A linear representation of the protein is displayed above the plot. PTMs with a score of 4 on iPTMnet[76] are marked at their respective positions, along with the detected cleavage sites. **d**, Fragment-level open search results examining the mass offsets corresponding to one or two phosphate groups. Data from three HCD acceleration voltages were combined before conducting the search. **e**, Scatter plot illustrating the mean number of phosphates between SPP1 residues and the nearest terminus (purple squares for the N terminus and pink circles for the C terminus) for full-length SPP1, as measured by nTDMS. Data are presented as mean ± s.d. from $n$ = 8 independent $MS^2$ spectra (acquired with different isolation windows and HCD acceleration voltages) of the same purified protein preparation. Previously observed phosphosites from iPTMnet[76] are displayed above the plot. The observed sequence ions do not span the central region of the protein.

release[61,62]. Given the importance of GABA homeostasis in maintaining healthy brain function, pharmacological modulation of GAT1 has been effectively utilized in the treatment of epilepsy[63] and suggested for the management of anxiety disorders[64]. Dysregulated GAT1 activity may directly contribute to such neurological and psychiatric disorders, although the molecular mechanisms (including PTMs) underlying such changes remain an active area of research.

To investigate PTM-based mechanisms of GAT1 regulation, we characterized the composition of human GAT1 using nTDMS. The native mass spectrum revealed a single major charge state distribution that was tentatively assigned as the monomeric transporter (Fig. 4b). Small populations of GAT1 bound to one or two phospholipids were also observed (approximately +760 and +1,400 Da). Upon expansion of the 18+ charge state, we identified three distinct peaks. The lowest intensity peak at $m/z$ 3,969.20 (18+, 71,427 Da) was consistent with the expected molecular weight of GAT1 with all three extracellular $N$-glycan sites occupied, N-terminal methionine cleaved, single acetylation and a disulfide bond (71,429 Da; Fig. 4c). Neither GAT1 methionine exclusion nor acetylation has been annotated in UniProt, yet the small hydrophobic residue at position 2 (Ala) makes the transporter a likely substrate of methionyl aminopeptidases and N-terminal acetyltransferases. GAT1 proteoforms with fewer or more than three glycans were not detected by native MS, indicating complete and exclusive occupancy at Asn176, Asn181 and Asn184. Such closely spaced glycosylation may influence GAT1 interactions with extracellular binding partners (Fig. 4c).

In addition to this low-mass proteoform, we observed two peaks of approximately equal abundance corresponding to +238 Da and +265.5 Da, the former constituting a mass difference that could be assigned to $S$-palmitoylation[65]. However, without biochemical interrogation, these imprecise mass shifts could also correspond to nonspecific HEPES adduction which shares the same nominal mass as palmitate[19]. To characterize the composition of these species without biochemical intervention, we selected the modified ions ($m/z$ 3983 ± 2.5) and irradiated them with 10.6-µm photons to generate fragments via stepped laser power infrared multiple photon dissociation (IRMPD)[16] (Fig. 4b and Supplementary Fig. 14).

precisION's fragment-level open search revealed a range of C-terminal modifications. Between 200 and 300 Da, we observed six significant offsets in close agreement with the mass of palmitate (1-mDa error) and stearate (5-mDa error), confirming that the mass shifts of +238 and +265.5 Da correspond to the lipidated transporter (Fig. 4d). By comparing the position of lipidated and unmodified sequence ions along the length of GAT1, we localized the fatty acids onto a single short stretch of residues containing two plausible modification sites, Cys493 and Cys499 (Fig. 4e). Intriguingly, the mass offset scan suggested the presence of distinct palmitate and stearate lipids with varied degrees of unsaturation. Close inspection of sequence ion envelopes uncovered 16:0, 18:1 and 18:0 fatty acid modifications—the latter differ by 2 Da (Fig. 4b). Fitting spectra with theoretical isotopic envelopes enabled us to quantify each lipid modification (Fig. 4f and Supplementary Figs. 15 and 16). We found that their abundances differ from the free fatty acid profile of the HEK293 GNTI$^{-/-}$ cells used for expression (Fig. 4f and Extended Data Fig. 9). The diversity of $S$-acylation[66] (palmitoylation, oleoylation and stearoylation) is shaped by both the availability of metabolic precursors and substrate specificity of the protein lipidation machinery. In the case of GAT1, conjugation of saturated 16- and 18-carbon fatty acids was preferred, with palmitoylation being the predominant modification despite its lower free fatty acid substrate availability.

To elucidate the structural effects of these lipid modifications on GAT1, we re-examined recent high-resolution electron cryo-microscopy (cryoEM) data that resolved binding of GABA and other ligands to the human transporter[62]. No lipidation was modeled in the reported structures. In the map of the GAT1–ligand complex, we observed density extending from Cys493 into a pocket formed at the junction of transmembrane domains 2, 7 and 11. The palmitate group identified using nTDMS could be modeled into this unassigned density (Fig. 4g). No

such density was observed at Cys499, enabling the combined MS and cryoEM datasets to unambiguously define the GAT1 modification site. We compared our updated model of GAT1 with recent structures of the porcine serotonin transporter SERT (*SLC6A4*) isolated from brain. In these structures, cholesterol was modeled into the same hydrophobic pocket that accommodates palmitate in GAT1, suggesting that lipidation may block cholesterol interactions[67] (Extended Data Fig. 10). Additionally, we identified density adjacent to palmitate in GAT1 that resembled a bound phosphatidylethanolamine (PtdEtn) lipid (Fig. 4g). This phospholipid was observed to form extensive contacts with the palmitate group, implying a potential structural role for this modification in the native membrane.

## Discussion

The development of precisION marks a step change in the analysis of modified protein complexes through nTDMS. Purpose-built to facilitate deep, data-driven analyses, this framework maximizes insights into proteoform diversity. Central to its success is the accuracy of fragment mass measurements, which underpin the confident elucidation and localization of PTMs. A particularly distinctive feature of precisION is its ability to leverage robust sets of internal fragments to examine protein regions not covered by canonical sequence ions. This innovative, statistically controlled approach broadens the scope and enhances the fidelity of fragment ion assignments.

The ability of precisION to derive biological insights depends critically on the quality of input spectra. While high-quality nTDMS data can now be routinely obtained using commercial instrumentation, it remains critical to optimize spectral quality during acquisition. Instruments with low-resolution mass analyzers or limited fragmentation capabilities are less likely to yield the same depth of insight into proteoform diversity, primarily due to the reduced interpretability of the resulting data. Furthermore, precisION's current methodologies are not yet fully compatible with systems where fragmentation is confined to a small number of specific sites; in our and others' experience, directed cleavage occurs when there are no mobile charge carriers available on the precursor ions[68–70]. In such cases, experimental adjustments are necessary to confidently identify relevant fragment ions. These adjustments may include enhancing fragmentation through ion supercharging or employing multistage tandem MS techniques.

We selected distinct systems to demonstrate the capabilities of precisION in characterizing heterogeneous protein complexes which are challenging for established top-down approaches. Specifically, for ACE2, we revealed glycan fragmentation patterns that can now be applied as a universal search option for assigning glycoprotein nTDMS spectra. For the endogenous heterotetrameric PDE6 complex, we observed complete C-terminal geranylgeranylation of the β subunit, suggesting the importance of such lipidation for complex maturation and assembly. For osteopontin (SPP1), our untargeted search methods uncovered modified fragments, quantifying phosphorylation across the length of the intact protein and identifying previously undescribed truncations. Without precisION, such subtle yet significant changes in mass would be obscured by the cloak of heterogeneity, hindering our understanding of osteopontin's functional diversity and its implications in health and disease. For membrane proteins, precisION overcomes the challenges of unambiguously identifying covalent lipid modifications, either within EM density maps or using standard proteomics approaches[71]. This capability is likely to become increasingly important, given that of the ~2,900 membrane protein structures derived from human cell lines, only 38 have palmitoyl cysteine modeled into the density[72]. Variation in lipid modifications (for example, as a function of disease state) may alter membrane protein interactions and dynamics, driving changes in cellular phenotypes[66,73]. We successfully detected the diversity of lipidation on GAT1 from a human cell line. Since these modifications likely direct proteins to specific membrane environments, this presents an exciting opportunity to link changes in protein lipidation with perturbations in membrane lipid binding.

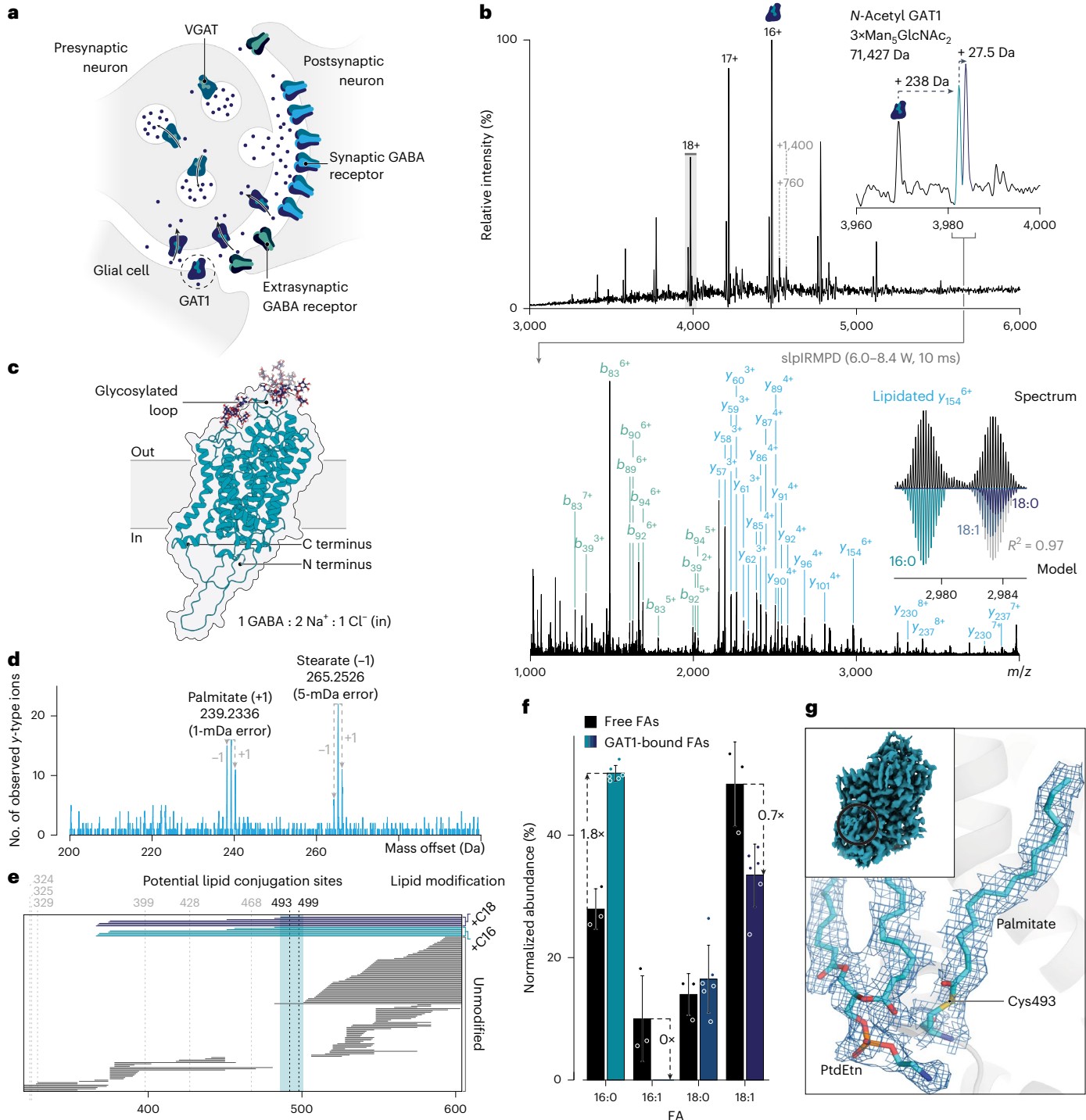

**Fig. 4 | The human GABA transporter GAT1 is covalently lipidated to fill an interaction site within the plasma membrane. a**, Schematic illustrating the action of GAT1 at synapses. GABA is released into the synaptic cleft where it activates ionotropic and metabotropic GABA receptors. After its release, GABA is transported into surrounding glial cells and the presynaptic neuron by GAT1. The neurotransmitter is then packaged back into vesicles by the vesicular GABA transporter (VGAT). **b**, Mass spectrum of GAT1 (400 mM ammonium acetate, 2 × CMC DDM/CHS, pH 7.0). Post-translationally modified ions ($m/z$ 3,983 ± 2.5) were selected using the quadrupole and activated with infrared photons (stepped laser power IRMPD (slpIRMPD) 6.0–8.4 W, 10 ms). The annotated MS[2] spectrum generated at a laser output power of 7.2 W is displayed below. Inset is the $y_{154}{}^{6+}$ sequence ion envelope. Differentially lipidated forms of this ion were detected and modeled. **c**, AlphaFold database structure of human GAT1 with representative glycan structures from GlycoSHIELD[77]. **d**, Fragment-level open search results for the C terminus of GAT1. **e**, Fragment map illustrating the position of the unmodified (gray) and lipidated (blues) sequence ions along the C terminus of GAT1. Individual horizontal lines correspond to sequence ions assigned from the combined slpIRMPD dataset. Potential cysteine residues where lipid modifications may occur are highlighted above the plot. **f**, Bar chart illustrating the relative abundances of different fatty acids in HEK293 GNTI[−/−] cells (black) or conjugated to Cys493 (blues). Data are presented as mean ± s.d.; for protein-bound lipids, $n = 6$ independent isotopic envelope fits, and for free fatty acids, $n = 3$ independent lipid extractions. **g**, EM density map (PTMs and lipid density contoured at 3σ) of the modeled palmitoyl cysteine and PtdEtn lipid on human GAT1 (EMD-33674, PDB 7Y7Y). The palmitate group was modeled covalently on Cys493 and a PtdEtn lipid was modeled adjacent to this moiety. FA, fatty acid.

Looking forward, we envision that precisION will redefine the capabilities of nTDMS in biomolecular analysis beyond the offerings of proteomics, driving the discovery of post- and co-translational modifications that influence protein structure and interactions. Our methods also demonstrate potential beyond native protein analysis, extending to high-throughput top-down proteomics. The fragment-level open search will enable top-down *N*-glycoproteomics, uncovering how glycans and other labile modifications communicate with other PTMs to facilitate precise molecular control. Integrating the resulting catalog of modifications with complementary approaches will refine structural models and interpretations of density maps[74,75]. Critically, as an open-source and vendor-independent platform, precisION will be made freely available, complete with detailed instructions and example datasets (https://github.com/kanalstrahlen/precisION). By enabling comparative, high-resolution analyses of functional protein assemblies across distinct (patho)physiological states, precisION is poised to uncover the molecular determinants of disease and deepen our understanding of biomolecular regulation.

## Online content

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

## Methods

### Protein expression and purification

Endogenous bovine PDE6 was isolated from rod outer segment disc membranes by washing with isotonic buffer (20 mM HEPES pH 7.5, 100 mM NaCl, 1 mM DTT, 5 mM $MgCl_2$) before pelleting membranes via centrifugation (31,000$g$, 1 h, 4 °C) as previously described[19].

The human ACE2 N432KO ectodomain was transiently overexpressed in suspension-adapted HEK293 GNTI$^{-/-}$ cells and purified using immobilized metal affinity chromatography and gel filtration as previously described[46].

*Gallus gallus* avidin was obtained as a lyophilized powder following its purification from egg white (Calbiochem). The lyophilized protein was redissolved in 50% MeCN, 1% formic acid in water supplemented with 5 mM TCEP (10 µM) and heated at 50 °C for 15 min, then incubated at 25 °C for a further 1 h. The reduced protein was directly analyzed from this denaturing solution.

Recombinant human SPP1 (UniProt isoform 1) was obtained as a lyophilized powder following its purification from HEK293 cells (PeproTech). The manufacturer confirmed protein activity as determined by its ability to enhance the cell adhesion of murine B16-F1 cells, indicating similar function to endogenous SPP1. Before nTDMS analysis, the lyophilized protein was redissolved in water (1 mg ml$^{-1}$) and flash-frozen in liquid nitrogen, before storage at −80 °C as per the manufacturer's instructions. To deglycosylate SPP1, we incubated the protein with a cocktail of α2–3,6,8,9 Neuraminidase A (New England Biolabs), β1–4 Galactosidase S (New England Biolabs), β-*N*-Acetylhexosaminidase (New England Biolabs) and *O*-Glycosidase (New England Biolabs) in 50 mM sodium phosphate (pH 7.5) at 37 °C before buffer exchange into 1 M ammonium acetate, pH 7.0.

Human GAT1 was stably overexpressed in suspension-adapted HEK293S GNTI$^{-/-}$ cells (CRL-3022; ATCC) following lentiviral transduction[78]. Briefly, a synthetic double-stranded DNA fragment (gBlock; Integrated DNA Technologies) encoding human SLC6A1 (P30531) with C-terminal, TEV-cleavable Avi- and FLAG-tags was cloned into a lentiviral transfer plasmid (pHR-CMV-TetO2-3C-Twin-Strep_IRES-EmGFP), a gift from A. R. Aricescu and J. Elegheert (Addgene plasmid no. 113884), between the EcoRI and the XhoI sites using In-Fusion assembly (Takara Bio)[78]. The resulting plasmid sequence was confirmed by Sanger sequencing. Lentiviral particles were generated in HEK293T Lenti-X cells (632180; Takara Bio) following co-transfection with equal amounts of the transfer plasmid, lentiviral packaging plasmid (psPAX2; Addgene plasmid no. 12260; a gift from D. Trono) and lentiviral envelope plasmid (pMD2.G; Addgene plasmid no. 12259; a gift from D. Trono) using Lipofectamine 2000 (Thermo Fisher Scientific). After 3 d, the secreted virus was used to infect HEK293 GNTI$^{-/-}$ cells grown as a monolayer[78]. After approximately 3 d, cells positive for emGFP were enriched using fluorescence-activated cell sorting. These cells were adapted to suspension growth in FreeStyle media (Gibco) supplemented with 1% FBS. Upon reaching a density of ~4.0 × 10$^6$ cells per ml, cells were collected by centrifugation (500$g$), washed in phosphate-buffered saline and frozen in liquid nitrogen.

All purification procedures were carried out on ice with cold (~4 °C) buffers. Frozen cells from ~2 l of culture were thawed in HNG buffer (50 mM HEPES pH 8.0, 150 mM NaCl, 5% glycerol) supplemented with a protease inhibitor cocktail (cOmplete; Roche) and lysed with three passages through a microfluidizer (12,000 psi; Microfluidics M-110P). The lysate was cleared by centrifugation (20,000$g$, 20 min) before membranes were pelleted using ultracentrifugation (180,000$g$, 1 h). The pelleted membranes were resuspended in HNG buffer with 10% glycerol and flash-frozen in liquid nitrogen before storage at −80 °C. Membranes were thawed on ice and solubilized for 3 h after homogenization in HNG buffer supplemented with 1.5% (w/v) *n*-dodecyl-β-D-maltoside (DDM) and 0.15% cholesteryl hemisuccinate (CHS). The solubilized mixture was clarified by centrifugation (20,000$g$, 20 min) before FLAG-tagged GAT1 was isolated by FLAG affinity purification. Anti-DYKDDDDK magnetic agarose (0.4 ml of settled beads) was incubated with the

solubilized membranes for 1 h, before washing with HNG buffer supplemented with 2 × critical micelle concentration (CMC) DDM/CHS (5 × 5 ml). GAT1 was eluted with 1 mg ml$^{-1}$ DYKDDDDK peptide (Genscript) for 1 h. The eluant was concentrated and washed using a centrifugal filter (100-kDa molecular weight cut-off; Amicon Ultra) and digested with TEV protease overnight in HNG buffer supplemented with 2 × CMC DDM/CHS and 5 mM tris(2-carboxyethyl)phosphine. After digestion, the 6 × His-tagged TEV protease was removed from the mixture using immobilized metal affinity chromatography and the resulting solution analyzed by nTDMS.

### nTDMS

All proteins (except *G. gallus* avidin) were exchanged into 200–1,000 mM ammonium acetate, pH 7.0 using gel filtration (75-µl Zeba spin columns; Thermo Fisher Scientific) and loaded into gold-coated nanoelectrospray ionization (nESI) emitters fabricated in-house for native MS analysis. In the case of GAT1, the analyte solution was supplemented with 2 × CMC *n*-decyl-β-D-maltoside/CHS. nTDMS measurements were performed using a hybrid quadrupole–linear ion trap–Orbitrap mass spectrometer (Orbitrap Ascend Structural Biology Tribrid; Thermo Fisher Scientific) equipped with a static nESI source and a 60-W $CO_2$ laser (Firestar Ti60; Synrad) with its beam focused into the high-pressure region of the quadrupole–linear ion trap[16]. Native protein ions were typically generated by applying a 0.8–1.2-kV potential to the nESI emitter relative to the instrument's sampling interface (100–200 °C) and mild in-source activation (<60 V offset between the S lens and MP00) was used to enhance ion desolvation. In some cases, for example, large protein complexes and membrane proteins, more extensive in-source activation (up to a 250 V offset) was required for effective desolvation—full details can be found in the respective raw data files. Typically, such high activation offsets also necessitated the application of a retarding compensation potential (approximately 3 V) between L0 and MP0 to prevent the transmission of small solvent and detergent clusters[79]. To maximize the transmission of large folded protein ions, the instrument was operated in intact protein mode at either standard or high pressure (8 or 20 mTorr $N_2$ in the ion routing multipoles, respectively). Radiofrequency amplitudes were maximized to enhance radial trapping of large protein ions.

MS$^1$ spectra were acquired in the Orbitrap at a resolution of 17,500–60,000 at $m/z$ 200 with the enhanced Fourier transform enabled. Automatic gain control (target of 4 × 10$^4$ to 4 × 10$^5$ charges, maximum inject time of 500 ms) was used to control injection times and prevent notable space charging effects. Individual scans were averaged after the transients were transformed into the $m/z$ domain and, in some cases, subjected to minimal Gaussian smoothing. Data were analyzed manually.

For MS$^2$ spectra, an ensemble of ions within a given $m/z$ window was selected using the quadrupole and subjected to beam-type collisional-induced dissociation (HCD) in the front high-pressure multipole, or IRMPD in the high-pressure region of the linear ion trap. Isolation windows for each dataset are displayed in the main text figures. MS$^2$ spectra were acquired in the Orbitrap at a resolution of 240,000 at $m/z$ 200 with the enhanced Fourier transform enabled and a mass range of $m/z$ 500–4,000 or $m/z$ 1,000–4,000. Automatic gain control was again used to minimize space charging effects. For each spectrum, up to 1,000 transients were averaged to enhance the signal-to-noise ratio. This averaging was implemented through the Tune instrument control software (v.4.1.4223) by activating the averaging function (Σ symbol) and setting 'Scans to average' to 999. Averaged scans were saved as raw files once the signal-to-noise ratio was sufficient to resolve low-abundance envelopes, typically when noise levels decreased below 0.5% relative intensity. Spectra were converted to space-delimited text file format using vendor software (QualBrowser v.4.5.4747.0; Thermo Fisher Scientific) and directly processed using precisION.

## precisION

precisION's graphical user interface and algorithms are written in Python 3 and have been tested on systems with at least 8 GB of RAM (16+ GB recommended) running Windows 10 or later. Both the source code and portable executables are available for download alongside comprehensive documentation and tutorial files.

**Spectral deconvolution and isotopic envelope identification.** precisION can process profile mass spectra in tab- or space-delimited text file formats and has been tested on high-resolution Orbitrap (Thermo Fisher Scientific) and Fourier transform ion cyclotron resonance MS data (Bruker)[27]. Raw spectra are first deconvolved using one of two user-selected workflows: rapid or extensive. The rapid workflow identifies peaks using a continuous wavelet transformation[80], while the extensive workflow uses a modified Richardson–Lucy deconvolution algorithm. This latter approach is similar to UniDec[81] and UniDecNMR[82], but the method has been modified to iteratively deconvolve windows of width $m/z$ 100. Each window is deconvolved using a Gaussian point spread function with varied sigma to account for the $m/z$-dependent peak width typical of Fourier transform-based mass analyzers (Supplementary Fig. 1). Although the Richardson–Lucy-based method is more computationally intensive than the continuous wavelet transformation, it offers two distinct advantages for in-depth spectral analysis: (1) it more effectively identifies overlapping signals in congested regions of the spectrum, enhancing isotopic envelope selection and characterization; and (2) it reduces the effects of noise on peak shape, affording enhanced profile spectra (Supplementary Fig. 1).

After deconvolution, isotopic envelopes corresponding to protein fragment ions are identified using two established algorithms, THRASH[35,36] and TopFD[34]. These algorithms take markedly different approaches to isotopic envelope identification. THRASH identifies envelopes iteratively from profile mass spectra, calculating the charge state of the most abundant peak in the spectrum before evaluating the surrounding signal's fit to a theoretical 'averagine' envelope based on the calculated average mass and charge[35]. If the isotopic envelope satisfies a specified goodness-of-fit threshold, the peaks associated with the envelope are removed from the spectrum and the process continues. Because THRASH is iterative, a set of envelopes selected with a strict fit threshold is not necessarily a subset of those that would be selected with a more relaxed threshold (Extended Data Fig. 3). Thus, in precisION's extensive workflow, THRASH is applied four times with varied fitting thresholds.

In contrast, TopFD[34] extends the MS-Deconv[83] framework by using a graph-based approach to select the best set of isotopic envelopes from a centroided spectrum, rather than scoring envelopes individually. It offers two envelope scoring methods: MS-Deconv and EnvCNN[38]. precisION implements both scoring methods, as they tend to identify different sets of envelopes, maximizing envelope detection sensitivity.

The results from both deconvolution algorithms are clustered by mass, and several quantitative features are calculated for each isotopic envelope. These features include the: (1) signal-to-noise ratio, where the noise is defined as the modal peak intensity[83]; (2) custom fit score; (3) interference score, which describes the proportion of signal within the span of the fragment ion's isotopic envelope that can be assigned to the fragment ion; (4) fraction of missing peaks; (5) mass errors for individual isotopologues; (6) $\chi^2$ goodness-of-fit test statistic and associated $P$ value; and (7) Pearson correlation coefficient and associated $P$ value, comparing the theoretical and observed isotopologue intensities.

**Envelope filtering.** precisION's approach to isotopic envelope identification aims to maximize the identification of true fragment ion envelopes within a mass spectrum. However, this approach can also lead to the detection of many artifactual envelopes that do not correspond to actual fragment ions. Such false identifications are especially pronounced during the analysis of large (membrane) protein complexes, where extensive envelope overlap and lower signal intensities greatly increase spectral complexity.

The inclusion of false isotopic envelopes in fragment ion mass lists introduces noise into downstream analyses including de novo sequencing, open database searches, fragment assignment and fragment-level open searches (Supplementary Fig. 7). However, in an nTDMS experiment where fragment generation is often low, discarding genuine envelopes through overly stringent filtering can hinder proteoform identification and characterization. Thus, we identified the need for an algorithm that can conservatively remove artifactual isotopic envelopes without discarding true ions.

Our initial attempts to manually filter envelope lists revealed that defined cut-offs applied to simple scoring measures such as the fit score, signal-to-noise ratio or interference score were ineffective due to the extensive overlap between real and artifactual envelopes (Extended Data Fig. 3). Consequently, precisION incorporates a supervised machine learning algorithm for isotopic envelope classification, designed to uncover patterns in these complex data. Unlike previous machine learning-based approaches for envelope classification[37], which optimized the $F_1$ score (the harmonic mean of the precision and recall) and aimed to create generalizable models, we focused on minimizing the false negative rate and developed a pipeline to create spectrum-specific classifiers.

We developed a workflow allowing models to be easily trained on individual spectra, ensuring operator- and spectrum-specific optimization. Briefly, users manually evaluate a representative set of isotopic envelopes from the spectrum (~100–200), generating a labeled dataset that is used to train both a logistic regression model and a gradient boosting model (for a list of training features, see Supplementary Table 1). Each model undergoes hyperparameter optimization to maximize recall, and the trained models are then applied to evaluate the entire set of envelopes—often numbering in the thousands. To avoid false negatives, a stringent voting scheme is implemented, whereby envelopes are only discarded if both models classify them as false. This conservative approach lowers the risk of discarding genuine isotopic envelopes, which we consider more harmful than maintaining a small population of false positives.

Because the models are trained on individual spectra, this method is highly adaptable and can be applied to data from different instruments and biomolecular systems. Additionally, trained models can be saved and reused to classify related datasets, particularly in cases in which there are a number of spectra of similar quality.

**Protein identification.** precisION implements two complementary methods to enable protein isoform-level identifications from nTDMS spectra. Importantly, neither method requires extensive sequence coverage or an intact subunit mass. Unlike many tools used in denaturing top-down mass spectrometry, precisION does not attempt to identify exact proteoforms through its search. Instead, it detects protein modifications during downstream analyses with the fragment assignment module.

The first algorithm uses a graph-based representation of the spectrum to identify sequence tags de novo which can then be matched to specific isoforms[41]. This approach is particularly effective for systems where fragmentation is driven by structural features, such as disordered and surface-accessible regions[40] or α-helices[16]. Long sequence tags (typically >8 residues) generally yield confident identifications, while shorter tags can be used to filter putative candidates.

The second algorithm mirrors open search strategies commonly used in bottom-up proteomics[42]. It generates theoretical fragments ($b/y$-type or $c/z$-type ions) from a library of mature protein isoforms and their unprocessed precursors, matching these fragments to the observed data. Unlike most open search methods, this library is not filtered by intact mass before fragment matching. Thus, to reduce the search space, side chain modifications are not considered during the

search. To assist in the interpretation of the search results, precisION calculates a native fragmentation propensity score which effectively ranks proteoform–spectrum matches in nTDMS database searches[39]. The open search method is most effective when fragmentation is driven by sequence (for example, fragmentation often occurs at the C terminus of aspartic acid residues[70]).

**Fragment-level open search.** precisION's fragment-level open search is designed to identify sets of sequence ions that share a common modification. These modifications can be regulated (for example, sequence variants, PTMs), spontaneous (for example, oxidation) or a consequence of gas-phase ion reactivity (for example, neutral loss or secondary fragmentation to form internal fragments). The main objective of the fragment-level open search is to identify sets of sequence ions with a common modification (in the form of a mass offset, $\Delta m$) that are larger than would be expected given random matching.

To identify sets of sequence ions all carrying a small modification ($\Delta m < \pm 1,500$ Da), precisION examines a range of mass offsets within a user-selected interval (Extended Data Fig. 4). For each mass offset, the observed ions are compared with the theoretical sequence ions shifted by that particular offset, and the matching ions are counted. The mass offset is then changed, and the process is repeated iteratively across the entire range of mass offsets defined by the user. The spacing between mass offsets is determined based on the mass tolerance, ensuring that the offset domain is oversampled (that is, using a sliding-window approach).

The probability of observing a set of $k$ matched ions at a specific offset is modeled by a Poisson distribution, where the expected value ($\lambda$) is the mean number of matches across the full offset range. Since true sets of modified ions are rare, $\lambda$ approximates the expected number of matches if there were no modified sequence ions in the spectrum. The $E$ value for observing $k$ matches at a given offset is then calculated as:

$$E \text{ value} = \frac{\lambda^k e^{-\lambda}}{k!} \times n \quad (1)$$

where $n$ is the total number of offsets evaluated during the search.

Significant mass offsets are interpreted as real modifications present on a set of fragment ions. Given the precision and accuracy of the mass measurement (mass errors less than 0.05 Da), these modifications can be assigned with relatively high confidence (Extended Data Fig. 4).

For larger modifications, such as secondary fragmentation events, the quasi-continuous sliding-window approach becomes infeasible. In these cases, precisION evaluates multiple specific notches that correspond to potential modifications[45]. Despite this difference in approach, the statistical framework for testing remains consistent. Specific applications of the multinotch search are described below.

**Termini identification.** Before the assignment of sequence ions, the protein's native termini must be identified. Termini may be predicted using databases and sequence analysis, taking into account features such as methionine exclusion, N-terminal acetylation or signal peptide cleavage. However, in some cases, proteins can be cleaved at unusual sites which are post- or co-translationally modified. To identify these atypical cleavage events, precisION conducts a multinotch search[45], where each notch is selected to correspond to successive single residue cleavages. Notches that correspond to a high number of matches are likely to indicate the true termini of the proteoform. To accommodate N- or C-terminal modifications, users can add defined PTMs to each terminus based on their specific hypotheses.

**Terminal sequence ion assignment.** Once the protein's termini and any N-/C-terminal modifications have been identified, precisION assigns sequence ions through a hierarchical, semi-supervised scheme that minimizes false ion discoveries. At energies just above

the threshold for backbone fragmentation, protonated terminal fragments are typically the most abundant ions in native top-down mass spectra[84,85]. Accordingly, precisION first attempts to assign terminal fragments without side chain modifications.

Given the tendency for mass analyzers to drift, initial matching is typically performed with a low-stringency mass tolerance (for example, 10 ppm). For all putative assignments (based on mass), theoretical ion envelopes are generated using BRAIN[86] and fit to the data. The goodness-of-fit is calculated, and if both the mass error and fitting score satisfy user-defined thresholds, the ion is assigned automatically. If either threshold is not satisfied, the user is prompted to manually evaluate the assignment. To prevent repeated matching, each assigned fragment ion is removed from the list of envelopes in a similar manner to TopMPI[87], a recently reported approach for the analysis of chimeric intact protein fragmentation spectra that may facilitate top-down data-independent acquisition MS[88].

After assigning the first set of unmodified ions, precisION considers potential variants of these ions, specifically dehydrated, deamidated and sodiated forms. In this phase, precisION employs finer matching tolerances (for example, 3 ppm), which are permitted by searching for ions with defined mass differences from the assigned ions, correcting for systematic mass errors. This cycle iterates multiple times until no further products can be assigned, ensuring that the major side products of fragmentation upon vibrational activation are assigned without unnecessarily widening the search space.

Following the assignment of terminal fragments, precisION recalibrates the spectrum using a linear function to minimize errors in the $m/z$ domain (Supplementary Fig. 3). Recalibration corrects for drifts in calibration across the spectrum, enabling the use of stringent mass thresholds in later analyses. Assignments that fall outside a user-selected fine mass tolerance are removed after calibration.

**Internal sequence ion assignment.** Upon extensive activation, sequence ions can undergo secondary fragmentation to form internal fragments. These are a special class of sequence ions that do not contain the protein's native N or C terminus[89]. Internal fragments can provide useful sequence information across regions of the protein not covered by terminal fragments, but reliably assigning them using mass and fit alone has proven challenging due to the dense array of theoretical ions in the mass domain[90–92].

To minimize false internal fragment assignments (Extended Data Fig. 6 and Supplementary Fig. 6), precisION employs a shared terminus filter during searching. It detects sets of internal fragments that share either an N- or C-terminal fragmentation site. Due to the directed nature of fragmentation in nTDMS, most internal fragments will belong to one of these sets, where the shared terminal site corresponds to a high-propensity fragmentation site (Supplementary Fig. 17). To identify such sets, precisION conducts a multinotch search fragment-level open search, where the detected 'modifications' correspond to the common loss of a peptide backbone fragment (Supplementary Fig. 5). Significant offsets are selected, and internal fragments are assigned from these sets as described above. For degenerate assignments (that is, observed ions that could be matched to multiple theoretical internal fragments), precisION prioritizes assignments with higher likelihoods as previously described[93].

**Modification discovery.** Once the internal fragment search is complete, precisION will have effectively assigned all unmodified sequence ions, leaving a list of envelopes enriched in modified fragments. Additionally, the mass errors of these ions will be greatly reduced following recalibration. At this stage, precisION can search for sets of modified sequence ions using the quasi-continuous fragment-level open search, as described above.

Once significant mass offsets are identified, these offsets should ideally be assigned to known modifications or variants with defined

chemical formulas. Mass errors are typically less than 0.05 Da, and often less than 0.01 Da, enabling confident assignment. However, identifying the exact molecular formula of such modifications can be challenging due to the possibility for multiple or complex modifications[94]. To assist users in assigning offsets, precisION can map modified ions onto the protein sequence to identify the modified residue and directly search UniMod for single modifications and variants that match the observations. Alternatively, molecular formula calculators, such as those provided in vendor software (QualBrowser; Thermo Fisher Scientific), can be used to assist in assignment.

Significant modifications can subsequently be mapped onto the protein sequence by precisION, enabling their localization and quantification. This is achieved by considering the offset (expressed as a chemical formula or mass shift) as a fixed modification on sequence ions.

**Settings.** In this study, fragment ions were matched to theoretical ions in precisION v.0.0.3 with a mass tolerance of 3 ppm. Fragments with a goodness-of-fit score above 0.75 were automatically assigned while all others were manually evaluated (Supplementary Fig. 18). For all fragment-level open searches, offsets with an $E$ value less than 0.01 were considered for assignment.

**Additional capabilities.** In addition to the core modules highlighted here, precisION includes powerful tools for data visualization, assessment of assignment quality (Supplementary Fig. 19 and Supplementary Note 2) and targeted proteoform analysis (in a similar manner to refs. 95–98). These capabilities are particularly valuable for localizing protein modifications and identifying structural features that influence protein fragmentation. Further details regarding these methods and additional information concerning the algorithms described above can be found in precisION's documentation.

### Ion-level FDR estimation
**Internal fragments.** To estimate the internal fragment FDR, we employed a target–decoy-like approach. We conducted a multinotch fragment-level open search on the ACE2 sceHCD dataset to determine the number of 'target' internal fragment matches, as illustrated in Extended Data Fig. 5a (upper blue trace). To calculate the number of matches without applying the shared termini filter, we summed the number of matches from all notches. In contrast, to calculate the number of matches with the shared termini filter, we summed only matches from notches deemed statistically significant ($E$ value < 0.01).

To create decoy datasets, we applied decoy mass offsets to each of the theoretical ions and repeated the multinotch search (Extended Data Fig. 5a, upper gray trace). We used four random decoy mass offsets corresponding to the molecular formulas $H_5$, $H_5O_2$, $H_7O_9$ and $H_9O_3$. The FDR without the shared termini filter enabled was calculated as the average ratio of decoy matches to target matches across all of the notches. In comparison, the FDR with the shared termini filter enabled was determined by first calculating the average number of matches per notch in the decoy analyses. The expected number of false matches for the target dataset was then calculated as the product of this value and the number of significant notches selected for assignment in the target dataset.

To validate these calculations, we compared our results with those from the established algorithm, TDValidator (run through ProSight Native v.1.0.24038)[95,99]. TDValidator estimates the internal fragment FDR by shuffling the protein sequence and calculating the number of internal fragment matches. Results from 300 shuffled sequences were averaged. The FDR reported by TDValidator followed a similar trend to our calculations but was consistently lower, likely due to an additional isotopic envelope fitting step during fragment matching (Extended Data Fig. 6). We also validated these calculations experimentally, using multistage MS measurements to sequence a selection of internal fragments observed upon high-energy activation of SPP1

(Supplementary Fig. 20). Putative internal fragments were reisolated and subjected to a further round of HCD (variable acceleration voltage based on precursor $m/z$), before mass analysis in the linear ion trap. Ion assignments were confirmed via de novo sequencing and spectral alignment[93] (Supplementary Fig. 20). The results from these experiments were consistent with our estimated ion-level FDR of ~5%.

**Overall ion-level FDR.** We also estimated the full fragment FDR for the ACE2 HCD 150-V dataset. We used a similar approach to that described above, where the product of the expected number of matches at a random offset and the number of significant offsets considered during assignment was used to estimate the number of false matches at each stage of assignment. In this case, the overall ion-level FDR was found to be 3.3%. We expect this low rate of false assignments to be reproducible across typical datasets. It is also important to note that the modification-level FDR is expected to be much smaller than this due to the strict thresholds used during the identification of significant mass offsets.

### PTM quantification
We quantified the abundance of different modifications from $MS^2$ spectra using precisION's Proteoform analysis module. This module directly fits theoretical fragment ion envelopes to the raw mass spectrum after peak picking, enabling fragment assignment and relative quantification without deconvolution. Care was taken to ensure that ions compared during quantification originated from fragmentation at the same site. This approach assumes that the modification does not notably alter fragmentation efficiency at that site—a reasonable assumption given the small modifications examined here. In all cases, intensities were normalized by ion charge. Further details relevant to each instance of quantification are provided below.

**Truncation of human ACE2.** We used UniDec v.7.0.2 (ref. 81) to quantify the abundance of each truncated proteoform in the $MS^1$ spectrum. The charge state envelopes corresponding to the ACE2 monomer were deconvolved, and the peaks in the resulting mass distribution were numerically integrated to quantify the abundance of the different truncated forms. To quantify each species from the $MS^2$ data, we analyzed a set of four $y$-type ions arising from cleavage at the same site ($y_{16}^{1+}, y_{17}^{1+}, y_{18}^{1+}, y_{19}^{1+}$). Mean values from the HCD 180-V and HCD 200-V datasets are presented.

**Phosphorylation.** We acquired HCD MS/MS spectra of SPP1 with varied isolation windows and HCD acceleration voltages (Supplementary Table 2). For each spectrum we searched for $b$-/$y$-type sequence ions generated from SPP1[17–314] with 0, 1, 2 or 3 phosphate groups. For each spectrum-specific set of differentially phosphorylated sequence ions, we calculated the relative abundance of the different phospho-states. We then grouped the differentially phosphorylated isotopic envelopes from across the spectra, retaining groups only if the relevant sequence ion was observed in at least three spectra. We calculated the mean relative abundance of each phospho-state for each sequence ion (Supplementary Fig. 12), then determined the weighted average at each site. In cases in which a specific phospho-state was not detected in a particular spectrum, we assumed the abundance to be 0.

To confirm the results using HCD, we compared the data with an electron transfer dissociation with supplemental collisional activation spectrum generated by isolating a broad range of SPP1 ions ($m/z$ 3,600 ± 200) and reacting these ions with fluoranthene radical anions in the linear ion trap for 25 ms. The resulting charge-reduced protein products were dissociated using HCD (normalized collision energy 25%). At lower collision energies than 25%, we could not observe sufficient sequence ion signal. Analysis was repeated in a similar manner as described above, albeit with a single data point.

**Lipidation of GAT1.** We quantified the abundances of different lipid modifications on GAT1 by analyzing three sets of lipidated sequence ions with distinct charges ($y_{123}^{5+}$, $y_{154}^{6+}$, $y_{230}^{9+}$). Each of these sequence ions was quantified using two IRMPD MS/MS spectra generated by isolating different GAT1 charge states (+18 and +19). For each sequence ion, we simulated the respective isotopic envelope for the fragment with a 16:0, 18:1 or 18:0 lipid attached and convolved the resulting delta functions with a Gaussian point spread function. The intensities of the individual convolved envelopes were varied to fit their sum to the observed data by minimizing the sum of the squared residuals. Standard deviations were calculated from the variance between the abundances derived from individual fits.

To derive the abundances of free fatty acids, we isolated lipids from the HEK293 cells used for GAT1 expression ($n = 3$) with the Bligh and Dyer method[100]. The chloroform phase was isolated, diluted twofold with methanol and analyzed by negative mode electrospray ionization MS using a hybrid quadrupole–linear ion trap–Orbitrap mass spectrometer (Orbitrap Eclipse; Thermo Fisher Scientific) equipped with a static nESI source. Fatty acids were assigned based on their intact mass with a 2-ppm tolerance. The response factors of all fatty acids were assumed to be equal.

### Structural modeling of lipidated GAT1
We used COOT v.0.9.8.92 (ref. 101) to model palmitoyl cysteine and a PtdEtn lipid into a high-resolution EM map of human GAT1 (EMD-33674, PDB 7Y7Y). Figures were prepared using Pymol v.3.0.0 (Schrodinger) and UCSF ChimeraX v.1.5 (ref. 102).

### Reporting summary
Further information on research design is available in the Nature Portfolio Reporting Summary linked to this article.

## Data availability
Raw data including raw spectra, lists of assignments and Alpha-Fold 3 outputs are available via figshare at https://doi.org/10.6084/m9.figshare.28009352 (ref. 103). Source data are provided with this paper.

## Code availability
precisION's source code and portable executables for Windows 10/11 are available via GitHub at https://github.com/kanalstrahlen/precisION, along with detailed instructions and example data. The version of the software that was reviewed (that is, a clone of the GitHub repository at the moment of review) is available via Zenodo at https://doi.org/10.5281/zenodo.16918486 (ref. 104).

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

## Acknowledgements

We thank T. Kirschbaum, S. A. S. Lawrence, S. A. Burnap, W. B. Struwe, J. E. P. Syka, J. D. Hinkle, C. Mullen, L. A. Baker, T. B. O'Hagan and E. Hsu for helpful discussion. We thank R. Hedley and V. Tsioligka for providing technical assistance with cell sorting at The Don Mason Facility of Flow Cytometry, Sir William Dunn School of Pathology, University of Oxford. T.J.E. is an E. P. Abraham Junior Research Fellow at Linacre College, Oxford. C.K. is a Postdoctoral Fellow of the German National Academy of Sciences Leopoldina (LPDS grant no. 2023-07, C.K.). C.A.L. is a Research Fellow at Wolfson College, Oxford. Work on 'A trans-omic platform to define molecular interactions underlying anhedonia at the blood–brain interface' is supported by Wellcome Leap as part of the Multi-Channel Psych Program (C.V.R., C.A.L. and T.J.E.). Work in the C.V.R. laboratory is also supported by a Medical Research Council program grant (no. MR/V028839/1, C.V.R.) and a Wellcome Trust Award (grant no. 221795/Z/20/Z, C.V.R.). Work in the J.L.P.B. laboratory is supported by the Wellcome LEAP Delta Tissue Program (J.L.P.B.). The funders had no role in study design, data collection and analysis, decision to publish or preparation of the manuscript.

## Author contributions

T.J.E., J.L.P.B., C.A.L. and C.V.R. acquired funding; J.L.B., T.J.E., C.A.L. and C.V.R. conceived the study; J.L.B. formulated, developed and implemented the algorithms; J.L.B., T.J.E., K.C.Z. and C.A.L. prepared protein samples and acquired mass spectra; J.L.B., C.K., H.S., F.I.B. and C.A.L. analyzed data; all authors provided input regarding the interpretation and presentation of results; J.L.B., C.A.L. and C.V.R. wrote the paper; all authors read and edited the paper.

## Competing interests

J.L.B., T.J.E., C.A.L. and C.V.R. are listed as inventors on a pending European patent application EP24190466 entitled 'Improved mass spectrometry methods', assigned to the University of Oxford, describing approaches for analyzing top-down mass spectra. C.V.R. is a cofounder of and scientific advisor at OMass Therapeutics. The other authors declare no competing interests.

## Additional information

**Extended data** is available for this paper at https://doi.org/10.1038/s41592-025-02846-5.

**Correspondence and requests for materials** should be addressed to Corinne A. Lutomski or Carol V. Robinson.

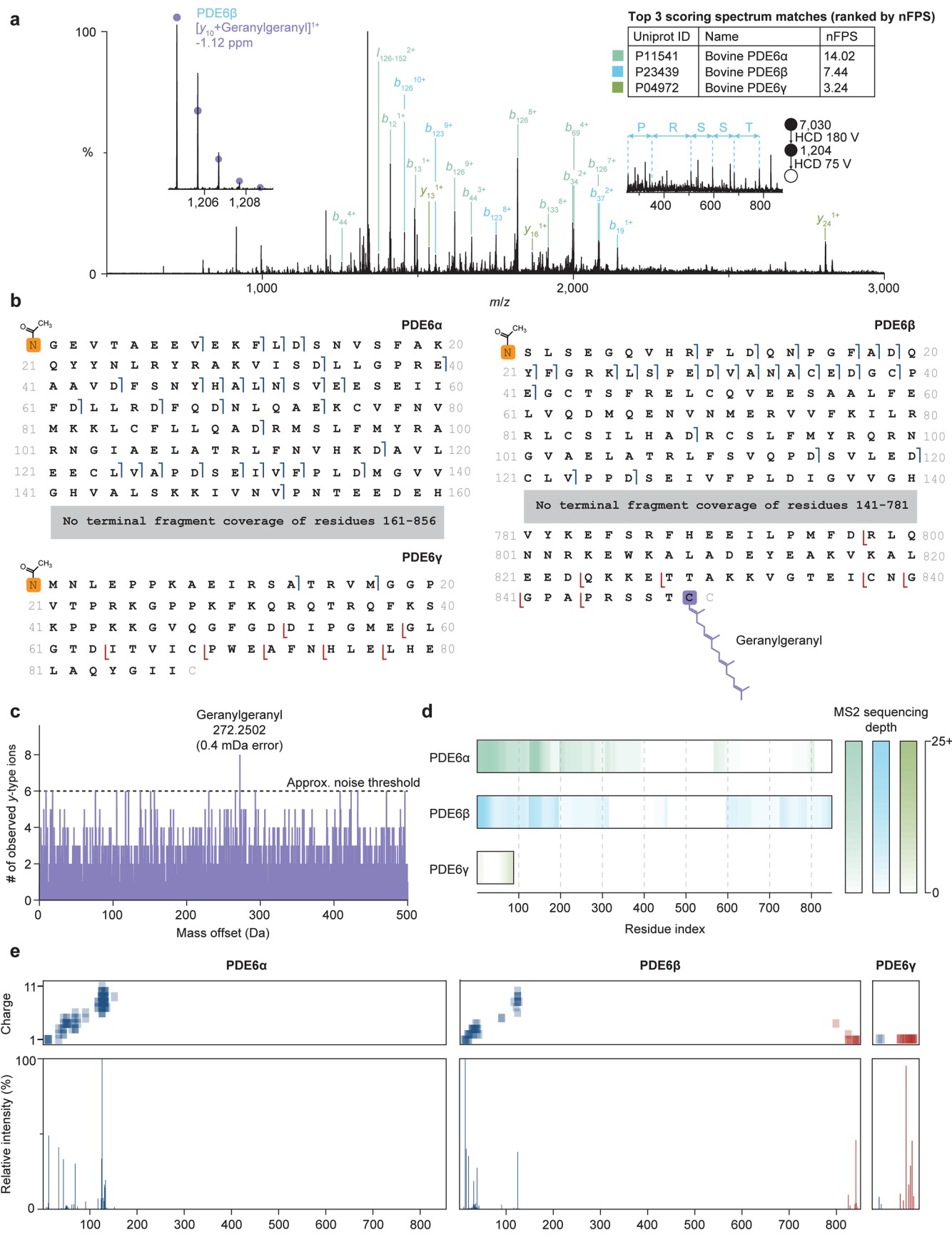

**Extended Data Fig. 1 | See next page for caption.**

**Extended Data Fig. 1 | Compositional analysis of endogenous PDE6 with precisION reveals insights into complex maturation. a**, Annotated native top-down mass spectrum of endogenous bovine PDE6 isolated from rod outer segment disc membranes (200 mM ammonium acetate, pH 7.0). The heterotetrameric complex (30+; $m/z$ 7250 ± 5; see Fig. 1a) was selected using the quadrupole and activated with infrared photons (IRMPD, 8.4 W, 5 ms). Inset are the isotopic envelope of the geranylgeranylated $y_{10}^{1+}$ sequence ion, the top three spectrum matches ranked by native fragmentation propensity score (nFPS), and a MS$^3$ spectrum supporting the geranylgeranylated $y_{10}^{1+}$ assignment. **b**, Sequence coverage maps for all three bovine PDE6 subunits highlighting observed terminal fragments. Regions without terminal coverage are noted. **c**, Fragment-level open search results for the C-terminus of PDE6β. A peak consistent with geranylgeranylation can be observed just above the noise. **d**, MS$^2$ sequencing depth per residue for PDE6α, PDE6β, and PDE6γ subunits, indicating how often each residue was observed within the set of sequence ions. **e**, Bar plots displaying total terminal fragment ion intensity for each cleavage site along the protein sequence, with projected scatter plots above indicating the charge states of the observed ions. $b$-type ions are displayed in blue while $y$-type ions are indicated in red.

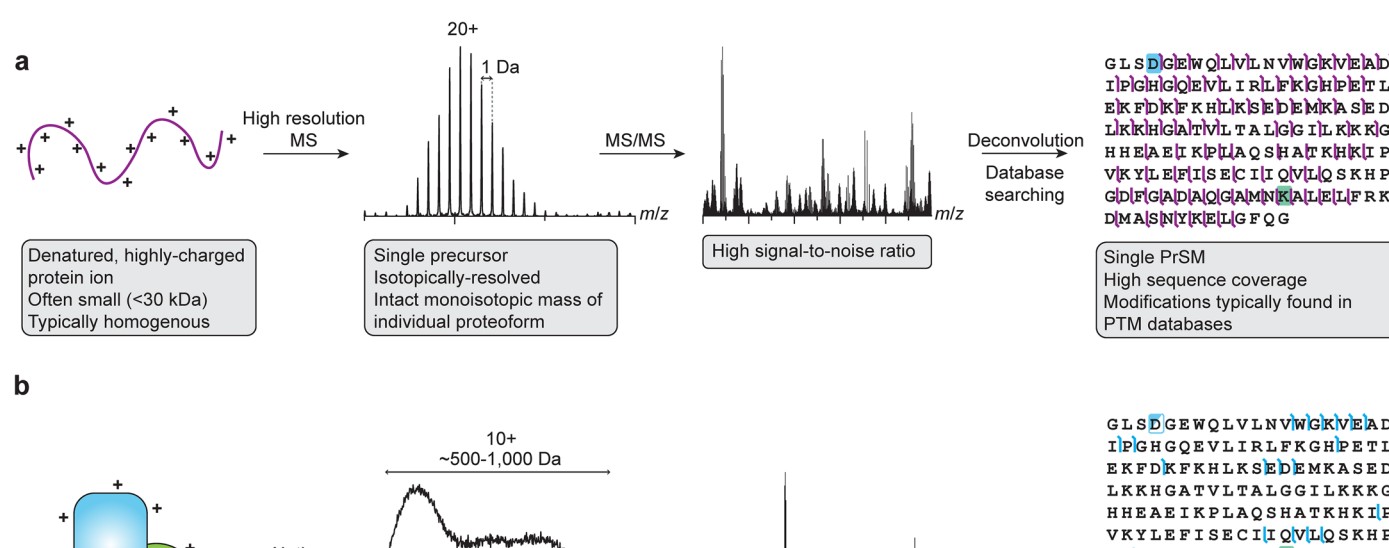

**Extended Data Fig. 2 | Denaturing and native top-down mass spectrometry measurements offer distinct data for proteoform identification. a**, Schematic overview of a typical denaturing top-down mass spectrometry measurement. Protein ions are generated *via* electrospray ionization from acidified mixtures of aqueous and organic solvents, often following online separation (for example, liquid chromatography). The resulting ions are typically monomeric, highly-charged, and often small and homogenous, facilitating analysis by high-resolution mass spectrometry. Fragmenting protein ions from denaturing solutions generally yields product ion spectra with high signal-to-noise ratios. The masses of the precursor and fragments can be used to identify the precursor ion by searching a library of theoretical proteoforms, such as through a traditional open search with a ±1–500 Da precursor tolerance. **b**, Schematic

overview of a native top-down mass spectrometry (nTDMS) measurement. Intact protein complexes are ionized from electrolyte solutions at physiological pH, typically using a static nanoelectrospray ion source. The resulting ions can comprise of multiple subunits (it is often not possible to dissociate individual subunits in the gas phase), each potentially bearing diverse modifications, affording a heterogenous set of unresolvable molecular species. Product ion spectra are typically of lower quality when compared to denaturing analyses, with reduced sequence coverage. Additionally, in targeted nTDMS studies, proteoforms with unexpected or uncommon post-translational modifications (PTMs) are frequently of particular interest—these PTMs are not represented in standard proteoform libraries. PrSM, proteoform–spectrum match.

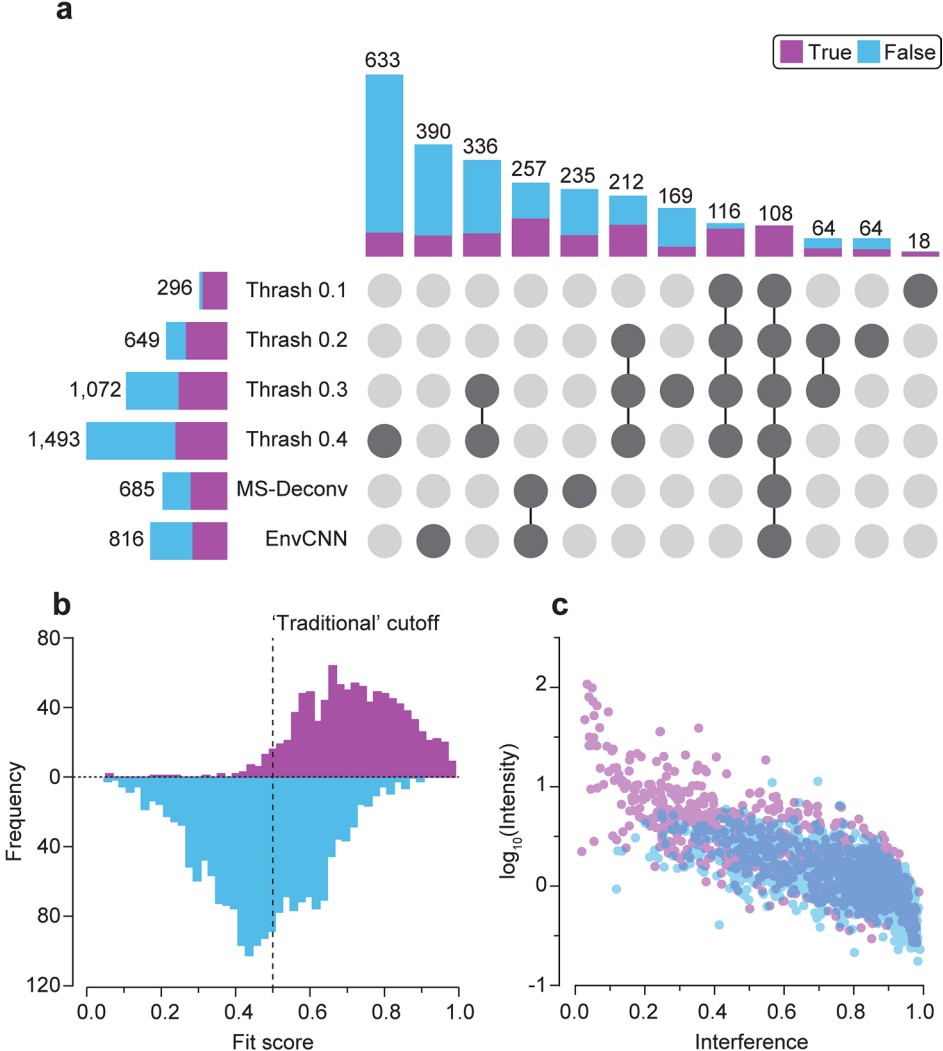

**Extended Data Fig. 3 | Real and artefactual isotopic envelopes detected by deconvolution algorithms cannot be readily discriminated using single scoring measures.** Summary of isotopic envelopes classified as true or false by precisION's supervised voting classifier. The envelopes were identified from a fragment spectrum of the ACE2 dimer activated using beam-type collision induced dissociation (HCD 130 V). **a**, UpSet plot illustrating the intersections between the isotopic envelopes detected by precisION's six deconvolution algorithms. The Thrash algorithm is executed with various scoring thresholds due to its iterative nature; stricter score requirements do not yield a subset of the envelopes identified with more lenient requirements. On the left side, horizontal bars represent the total number of envelopes each algorithm detected, with purple indicating true envelopes and blue indicating false ones. In the main plot, columns of filled circles show which algorithms detected the same sets of envelopes. Each column represents a unique combination of algorithms, and the bar above each column shows the number of envelopes consistently detected by that combination. Sets with less than 15 members were excluded from the plot. **b**, Frequency histograms showing the distribution of envelope fit scores for both true and false envelopes. Notably, many artefactual envelopes exhibit higher fit scores than true envelopes. Isotopic envelopes are often filtered by fit score in established workflows. **c**, Scatter plot depicting the distribution of intensity and interference scores for true and false envelopes. There is substantial overlap between the two classifications, illustrating the difficulty in distinguishing between real and artefactual envelopes.

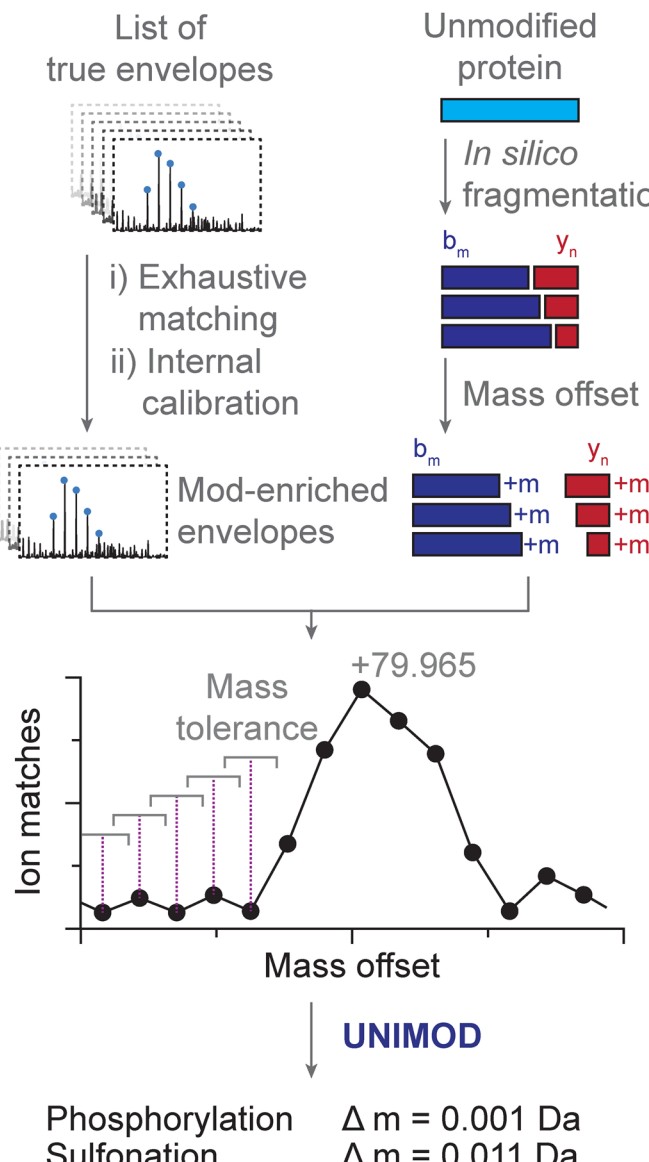

**Extended Data Fig. 4 | precisION's fragment-level open search can detect a diverse range of fragment ion modifications without constraint.** Schematic overview of the fragment-level open search. In a first pass search, observed ions are assigned to unmodified sequence ions. After this search, the residual unassigned ions are assumed to mainly consist of modified sequence ions. To identify these modifications, a mass offset is applied to each of the theoretical sequence ions, then the number of matches between the observed ions and the offset theoretical ions is counted. By scanning across a continuous range of mass offsets using a sliding window approach, the algorithm can identify offsets that produce a significantly high number of matches. Peaks in this scan indicate sets of sequence ions with a common modification. Here an example mass offset of +79.965 is shown to result in an increased number of matches compared to the background. Searching for this mass offset in UniMod would suggest it to correspond to a set of phosphorylated sequence ions.

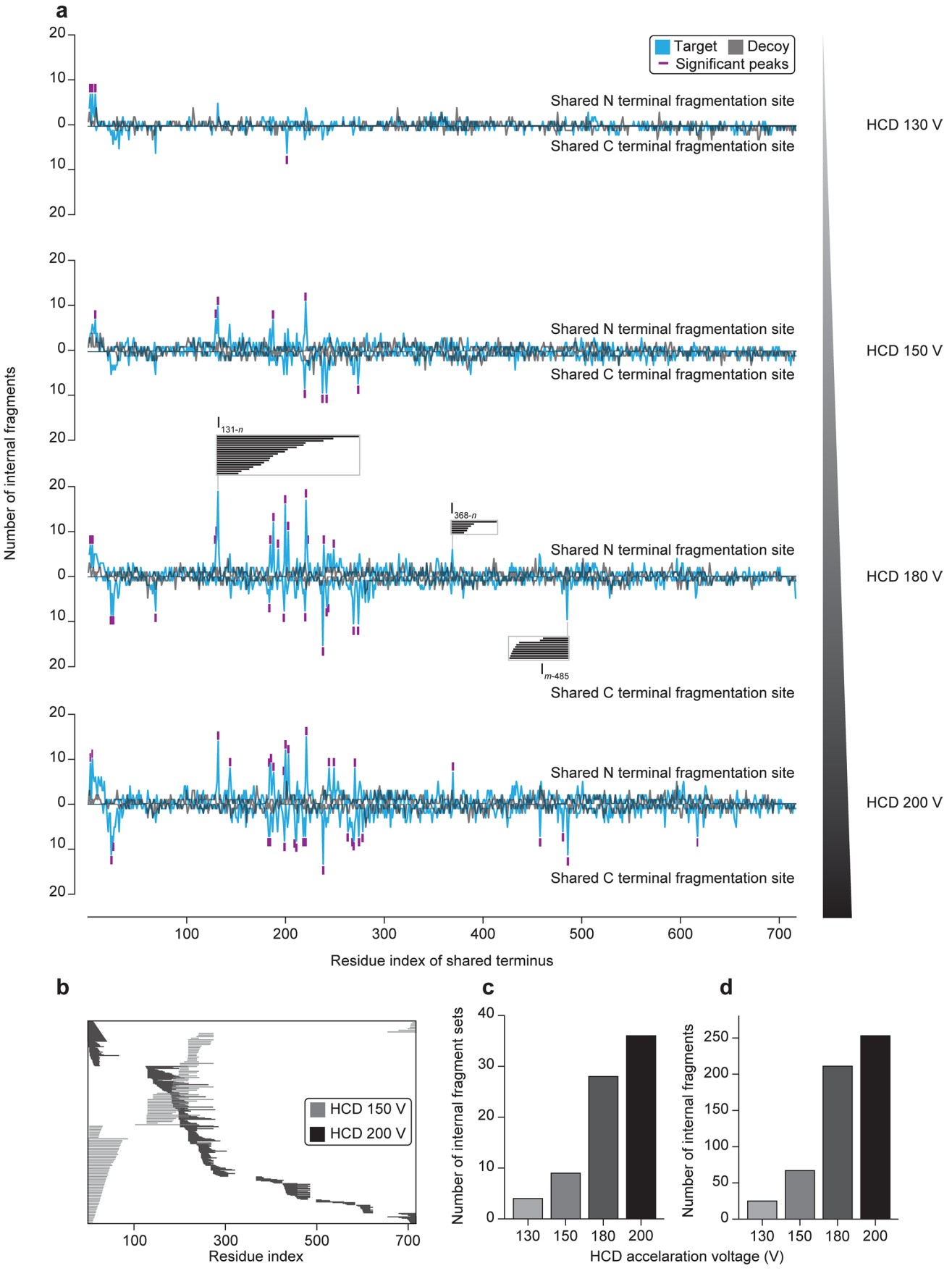

**Extended Data Fig. 5 | See next page for caption.**

**Extended Data Fig. 5 | Internal fragments generated by the collisional activation of native protein complexes share common terminal fragmentation sites. a**, Counts of internal fragments sharing specific N- or C- terminal fragmentation sites observed upon collisional activation of the dimeric ACE2 complex. Counts for each set of internal fragments sharing a N- (upper) or C-terminal (lower) fragmentation site are presented for the 'true' target set of theoretical ions (blue), as well as for an example set of decoy ions generated by adding the mass of acetate (42.0105 Da) to each theoretical ion (grey). Across all examined collision energies, there are sets of internal fragments with a shared fragmentation site that were larger in population than anticipated when assuming random matching. Sets of internal fragments with a shared fragmentation site that were deemed statistically significant (expectation value < 0.01) were selected for assignment (purple marks). **b**, Distribution of the assigned internal fragments along the length of ACE2. Horizontal lines represent individual sequence ions, with ions generated at HCD 150 V shown in grey and at HCD 200 V shown in black. Internal fragments are formed from localized areas, and are primarily generated by cleavage at high-propensity fragmentation sites (for example, D|P fragments). **c**, Number of statistically significant (expectation value < 0.01) sets of internal fragments with a common terminal fragmentation site identified at increasing HCD acceleration voltages. **d**, Number of internal fragments assigned at increasing HCD acceleration voltages.

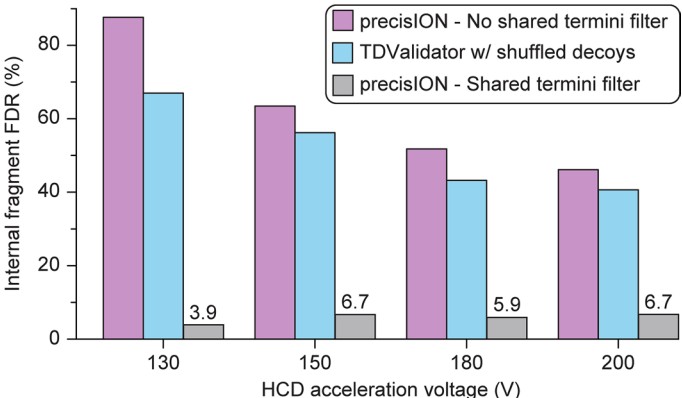

**Extended Data Fig. 6 | precisION's shared termini filter enables robust internal fragment assignment.** Fragment false discovery rate (FDR) for internal fragments generated upon collision activation of the ACE2 dimer (HCD 130–200 V). Without applying the shared termini filter, FDRs exceed 40%, whether using precisION or TDValidator (which additionally incorporates an envelope fitting score filter). By applying precisION's shared termini filter with an E-value threshold of 0.01, the FDR is reduced up to ~20-fold, even before further examination of poor envelope fits.

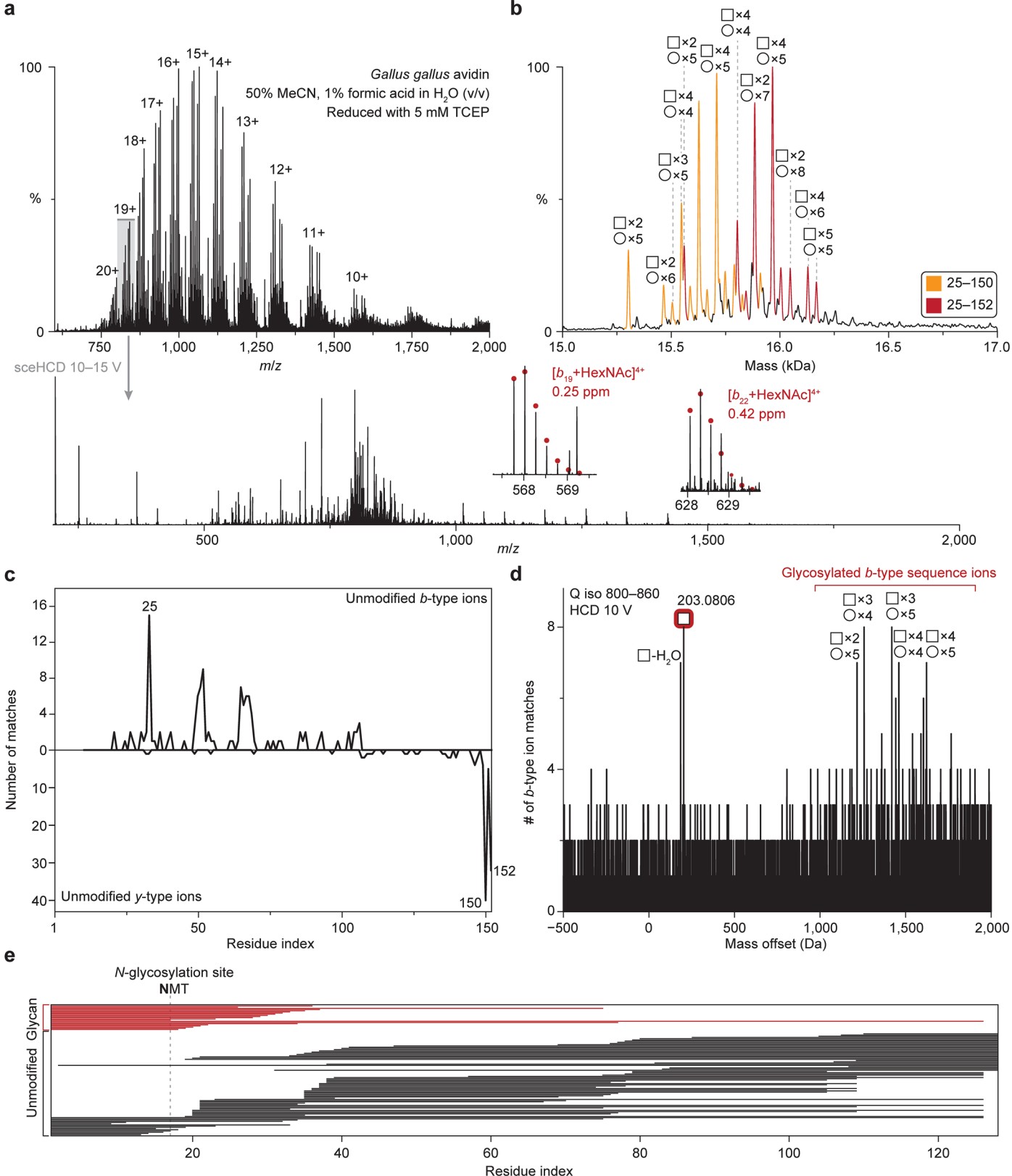

**Extended Data Fig. 7 | See next page for caption.**

**Extended Data Fig. 7 | N-glycans can be confidently identified and localized on intact, denatured proteins using top-down mass spectrometry with collisional activation. a**, Mass spectrum of *Gallus gallus* avidin acquired under denaturing and reducing conditions (50% MeCN, 1% formic acid in $H_2O$ (v/v), 5 mM TCEP). An envelope of highly-charged proteoforms was selected (19+; *m/z* 830 ± 30) with the quadrupole and activated using ion–neutral collisions (HCD 10-15 V) to yield abundant sequence ions. The resulting MS$^2$ spectrum generated at an HCD acceleration voltage of 12.5 V is displayed below. Inset are isotopic envelopes corresponding to sequence ions that have retained a HexNAc moiety throughout fragmentation. **b**, Deconvoluted mass spectrum illustrating the distribution of avidin proteoforms existing in the sample. All assignments were made on the basis of findings from top-down MS measurements. **c**, Multinotch fragment-level open search used to identify avidin termini. Signal peptide cleavage can be observed along with variable C-terminal truncation. **d**, Fragment-level open search used to identify N-terminal modifications on avidin. *b*-type ions retaining HexNAc or large glycans can be observed. These ions were used to inform compositional assignments at the MS$^1$ level. **e**, Fragment map illustrating the position of the fragmentation sites along the backbone of avidin. Individual horizontal lines correspond to sequence ions assigned from the combined HCD dataset. Sequence ions retaining glycan structures are displayed in red. *N*-glycan sequons (N-X-S/T) are highlighted with dashed lines.

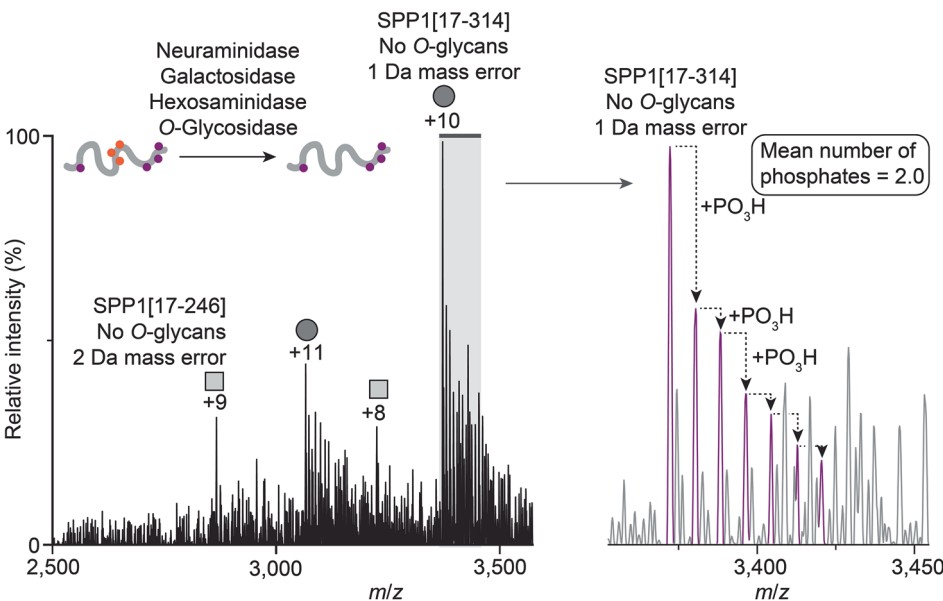

**Extended Data Fig. 8 | Deglycosylation of SPP1 confirms low protein phosphorylation.** Native mass spectrum of human SPP1 after treatment with a cocktail of glycosidases to completely remove all *O*-glycans. Two truncated proteoforms could be observed above the noise: SPP1[17-314] and SPP1[17-246]. Proteoforms with up to six phosphate groups could be observed for the full-length protein. However, the major stoichiometric form had no phosphate modifications. The average number of phosphates per protein molecule across the full protein population was calculated to be 2.0.

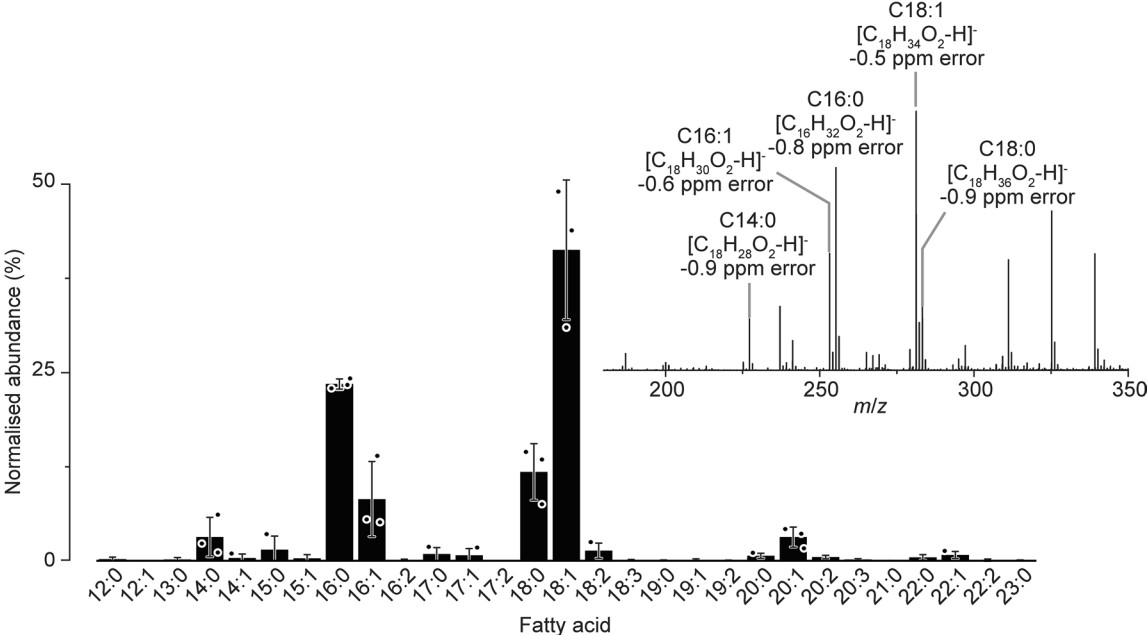

**Extended Data Fig. 9 | Free fatty acid profile of the cell line used for GAT1 expression.** Normalized abundance of free fatty acids in the HEK293 GNTI$^{-/-}$ cell line used for GAT1 expression. A representative negative mode electrospray ionization mass spectrum used for quantification is displayed as an inset. Fatty acids were observed as carboxylates and are notated as Cx:y where x is the chain length and y is the number of double bonds. Data are presented as the mean ± s.d. from n=3 independent lipid extractions. Individual data points greater than 1% are displayed.

**a**

hGAT1 (7Y7Y)

**b**

pSERT (8DE4)

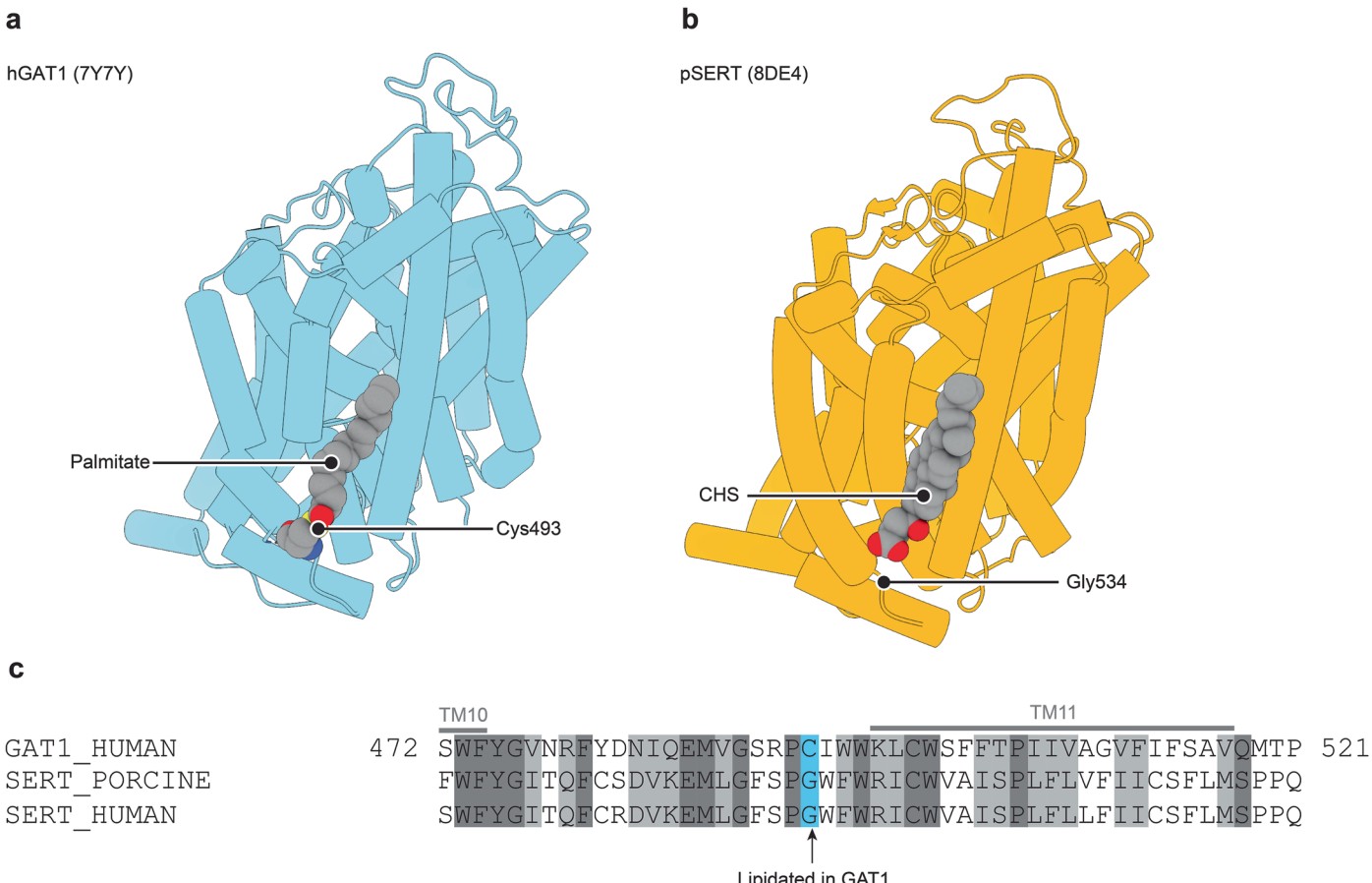

**c**

```
              TM10                                              TM11
GAT1_HUMAN    472 SWFYGVNRFYDNIQEMVGSRPCIWWKLCWSFFTPIIVAGVFIFSAVQMTP 521
SERT_PORCINE      FWFYGITQFCSDVKEMLGFSPGWFWRICWVAISPLFLVFIICSFLMSPPQ
SERT_HUMAN        SWFYGITQFCRDVKEMLGFSPGWFWRICWVAISPLFLLFIICSFLMSPPQ
                                       ↑
                               Lipidated in GAT1
```

**Extended Data Fig. 10 | GAT1 lipidation fills a potential lipid binding site by mimicking cholesterol. a**, Structure of human GAT1 (PDB ID 7Y7Y) with a palmitate moiety attached to Cys493 (grey). **b**, Structure of porcine SERT (PDB ID 8DE4) with a bound cholesterol hemisuccinate lipid (grey). **c**, Sequence alignment of human GAT1 and porcine and human SERT. The lipidated cysteine in GAT1 (Cys493) is not present in either SERT, preventing lipid conjugation at this site.

# Reporting Summary

## Statistics

For all statistical analyses, confirm that the following items are present in the figure legend, table legend, main text, or Methods section.

| n/a | Confirmed | |
|---|---|---|
| ☐ | ☒ | The exact sample size (*n*) for each experimental group/condition, given as a discrete number and unit of measurement |
| ☐ | ☒ | A statement on whether measurements were taken from distinct samples or whether the same sample was measured repeatedly |
| ☐ | ☒ | The statistical test(s) used AND whether they are one- or two-sided<br>*Only common tests should be described solely by name; describe more complex techniques in the Methods section.* |
| ☒ | ☐ | A description of all covariates tested |
| ☐ | ☒ | A description of any assumptions or corrections, such as tests of normality and adjustment for multiple comparisons |
| ☐ | ☒ | A full description of the statistical parameters including central tendency (e.g. means) or other basic estimates (e.g. regression coefficient) AND variation (e.g. standard deviation) or associated estimates of uncertainty (e.g. confidence intervals) |
| ☐ | ☒ | For null hypothesis testing, the test statistic (e.g. $F$, $t$, $r$) with confidence intervals, effect sizes, degrees of freedom and $P$ value noted<br>*Give P values as exact values whenever suitable.* |
| ☒ | ☐ | For Bayesian analysis, information on the choice of priors and Markov chain Monte Carlo settings |
| ☒ | ☐ | For hierarchical and complex designs, identification of the appropriate level for tests and full reporting of outcomes |
| ☐ | ☒ | Estimates of effect sizes (e.g. Cohen's *d*, Pearson's *r*), indicating how they were calculated |

*Our web collection on statistics for biologists contains articles on many of the points above.*

## Software and code

Policy information about availability of computer code

| Data collection | Orbitrap Ascend Tune Application v.4.1.4223, Orbitrap Eclipse Tune Application v.4.1.4223 |
|---|---|
| Data analysis | Thermo Xcalibur QualBrowser v.4.5.4747.0, precisION v.0.0.3 (v.0.2.0 for revisions), UniDec v.7.0.2, COOT v.0.9.8.92, Pymol v.3.0.0, UCSF ChimeraX v.1.5, Prosight Native v.1.0.24208, Python v.3.9.18, MagicPlot v.3.0.1 |

For manuscripts utilizing custom algorithms or software that are central to the research but not yet described in published literature, software must be made available to editors and reviewers. We strongly encourage code deposition in a community repository (e.g. GitHub). See the Nature Portfolio guidelines for submitting code & software for further information.

## Data

Policy information about availability of data

All manuscripts must include a data availability statement. This statement should provide the following information, where applicable:
- Accession codes, unique identifiers, or web links for publicly available datasets
- A description of any restrictions on data availability
- For clinical datasets or third party data, please ensure that the statement adheres to our policy

The data supporting the main conclusions of this study are presented in the Article and in the Supporting Information. Raw mass spectrometry data, assigned envelope lists, and structural models supporting the main text figures are available via Figshare (https://doi.org/10.6084/m9.figshare.28009352). Source data are

provided with the paper. precisION's source code and portable executables for Windows 10/11 are available via GitHub at https://github.com/kanalstrahlen/precisION along with detailed instruction and example data.

# Human research participants

Policy information about studies involving human research participants and Sex and Gender in Research.

| | |
|---|---|
| Reporting on sex and gender | N/A |
| Population characteristics | N/A |
| Recruitment | N/A |
| Ethics oversight | N/A |

Note that full information on the approval of the study protocol must also be provided in the manuscript.

# Field-specific reporting

Please select the one below that is the best fit for your research. If you are not sure, read the appropriate sections before making your selection.

☒ Life sciences   ☐ Behavioural & social sciences   ☐ Ecological, evolutionary & environmental sciences

For a reference copy of the document with all sections, see nature.com/documents/nr-reporting-summary-flat.pdf

# Life sciences study design

All studies must disclose on these points even when the disclosure is negative.

| | |
|---|---|
| Sample size | No sample size calculations were performed. |
| Data exclusions | No data were excluded from the analysis. |
| Replication | Quantitative measurements were performed in at least triplicate as described throughout the manuscript. |
| Randomization | No randomization was required. |
| Blinding | No blinding was necessary. |

# Reporting for specific materials, systems and methods

We require information from authors about some types of materials, experimental systems and methods used in many studies. Here, indicate whether each material, system or method listed is relevant to your study. If you are not sure if a list item applies to your research, read the appropriate section before selecting a response.

## Materials & experimental systems

| n/a | Involved in the study |
|---|---|
| ☒ | Antibodies |
| ☐ | Eukaryotic cell lines ☒ |
| ☒ | Palaeontology and archaeology |
| ☒ | Animals and other organisms |
| ☒ | Clinical data |
| ☒ | Dual use research of concern |

## Methods

| n/a | Involved in the study |
|---|---|
| ☒ | ChIP-seq |
| ☒ | Flow cytometry |
| ☒ | MRI-based neuroimaging |

## Eukaryotic cell lines

Policy information about cell lines and Sex and Gender in Research

| | |
|---|---|
| Cell line source(s) | HEK293S GNTI -/- cells were obtained commercially from ATCC (CRL-3022). Lenti-X 293T cells were obtained commercially from Takara Biosciences (632180). |

| Authentication | Commercially-sourced cell lines were used without further authentication. |
|---|---|
| Mycoplasma contamination | Cells were routinely tested for mycoplasma contamination (negative). |
| Commonly misidentified lines<br>(See ICLAC register) | N/A |

