## [Peer Review File · Nature Methods]

Uncovering hidden protein modifications with native top-down mass spectrometry

Corresponding Author: Professor Carol Robinson

A version of this paper was originally rejected for publication by Nature Methods, however that decision was reconsidered after appeal by the authors.

Version 0:

Decision Letter:

19th Dec 2024

Dear Professor Robinson,

Thank you for submitting your manuscript entitled "Uncovering hidden protein modifications with native top-down mass spectrometry". We have given the paper our careful consideration but we regret that we cannot publish it in Nature Methods.

It is Nature Methods' policy to decline a substantial proportion of manuscripts without peer-review, so that they may be sent elsewhere without delay. Decisions of this kind are made by the editors when it appears that papers are unlikely to succeed in the competition for limited space.

Among the considerations that arise at this stage are a manuscript's probable interest, level of methodological development and immediate practical relevance to a general readership. We do not doubt the technical quality of your work or that it will be of interest to others working in this area of protein research. However, I am sorry to say we do not think that the technical advances presented will have a sufficiently significant and immediate impact on a broader readership to justify publication in Nature Methods.

Although we cannot offer to publish your manuscript, I suggest that you consider Nature Communications as a suitable venue for this work. To transfer your manuscript, please use our manuscript transfer portal. You will not have to re-supply manuscript metadata and files, unless you wish to make modifications. For more information, please see our [manuscript transfer FAQ](http://www.nature.com/authors/author_resources/transfer_manuscripts.html?WT.mc_id=EMI_NPG_1511_AUTHORTRANSF&WT.ec_id=AUTHOR) page.

Nevertheless, thank you very much for giving us the opportunity to consider your manuscript. I am sorry that we cannot be more positive on this occasion and hope that you will promptly find a more appropriate forum for presenting your work.

Sincerely,
Arunima

Arunima Singh, Ph.D.
Senior Editor
Nature Methods

** For Nature Portfolio general information and news for authors, see <http://npg.nature.com/authors>

Version 1:

Decision Letter:

9th Apr 2025

Dear Carol,

Your Article, "Uncovering hidden protein modifications with native top-down mass spectrometry", has now been seen by 4 reviewers. As you will see from their comments below, although the reviewers find your work of considerable potential interest, they have raised a number of concerns. We are interested in the possibility of publishing your paper in Nature Methods, but would like to consider your response to these concerns before we reach a final decision on publication.

We therefore invite you to revise your manuscript to address these concerns. In particular, it would be important to highlight the utility and generalizability of the approach (with new demonstrations, if feasible) in addition to addressing all the other concerns raised by the reviewers.

Link Redacted

We hope to receive your revised paper within 8 weeks. If you cannot send it within this time, please let us know. In this event, we will still be happy to reconsider your paper at a later date so long as nothing similar has been accepted for publication at Nature Methods or published elsewhere.

OPEN SCIENCE REQUIREMENTS

REPORTING SUMMARY AND EDITORIAL POLICY CHECKLISTS

IMAGE INTEGRITY

When submitting the revised version of your manuscript, please pay close attention to our >Digital Image Integrity Guidelines and to the following points below:

EXTENDED DATA FIGURES

DATA AVAILABILITY

All novel DNA and RNA sequencing data, protein sequences, genetic polymorphisms, linked genotype and phenotype data, gene expression data, macromolecular structures, and proteomics data must be deposited in a publicly accessible database, and accession codes and associated hyperlinks must be provided in the "Data Availability" section.

CODE AVAILABILITY

Please include a "Code Availability" subsection in the Online Methods which details how your custom code is made available. Only in rare cases (where code is not central to the main conclusions of the paper) is the statement "available upon request" allowed (and reasons should be specified).

For more information on our code sharing policy and requirements, please see: <https://www.nature.com/nature-research/editorial-policies/reporting-standards#availability-of-computer-code>

MATERIALS AVAILABILITY

More details about our materials availability policy can be found at <https://www.nature.com/nature-portfolio/editorial->

SUPPLEMENTARY PROTOCOL

To help facilitate reproducibility and uptake of your method, we ask you to prepare a step-by-step Supplementary Protocol for the method described in this paper. We [encourage authors to share their step-by-step experimental protocols](https://www.nature.com/nature-research/editorial-policies/reporting-standards#protocols) on a protocol sharing platform of their choice and report the protocol DOI in the reference list. Nature Portfolio's protocols.io is a free-to-use and open resource for protocols; protocols deposited onto protocols.io are citable and can be linked from the published article. More details can be found at [protocols.io](https://www.protocols.io/help/publish-articles).

ORCID

Sincerely,
Arunima

Arunima Singh, Ph.D.
Senior Editor
Nature Methods

Reviewers' Comments:

Reviewer #1 (Remarks to the Author):

The authors present a computational method, precisION, to facilitate processing of native top-down mass spectra. They demonstrate their software using three proteins, one of which is a homodimer and the other two are monomeric. In each of the three cases, the samples contain a mix of proteoforms arising due to truncation, glycosylation, phosphorylation, etc. The ability of the precisION software to deal with complex fragmentation spectra is impressive and this tool will be highly useful to the community; however, I have a few comments the authors could consider.

The authors make the valid point in the introduction that approaches such as deglycosylation and in-solution disassembly sacrifice biological context and insight. Indeed, native (top-down) MS can allow the correlation of proteoform identity to the ability to form noncovalent assemblies. As such, I wonder if the three model systems chosen by the authors really showcase the full capabilities of precisION to provide biological insight, as only one of the models is a complex. Even in that case (a homodimer), it is not clear whether this dataset allows linking of proteoform identity to a tendency to engage in dimer formation. For example, I am curious whether the monomers in Figure 2b exhibit a different glycosylation pattern than the proteoforms incorporated in the dimer. The authors might also consider including data on a heterocomplex as a demonstration of the use of this software in an experiment in which native top-down MS provides clear added value.

Somewhat related to my previous point: As two of the proteins chosen by the authors to demonstrate precisION are monomers, I am curious to know whether the software can also tackle top-down spectra of denatured precursors, possibly with a broad isolation window. It is not obvious to me from the description of the software that this wouldn't be possible (although I suppose fragmentation might be a bit less directed and filtering internal fragments might be more difficult as a result), but as the majority of top-down MS is still carried out in denaturing mode, this seems worth testing to expand the potential user base of precisION. If the authors have tested this, or if it is fundamentally impossible to use this software with denatured precursors, it would be good to note this in the manuscript.

Minor points:

Top-down MS of a mixture of unresolved proteoforms, or of a complex consisting of different proteoforms can to an extent be seen as a special case of data-independent analysis and the challenges associated with the chimeric spectra generated in top-down DIA. While the precisION framework seems like a powerful new approach, there is considerable literature about this already, which the authors should at least briefly discuss and cite.

In the caption for Figure 1, the authors write that mass resolution for intact complexes is fairly low (10-500 Da). Do they really mean Da, or m/z? Similarly, they write that fragment ions are 'measured with high mass resolution and accuracy (typically <

0.02 Da)'. Does that value refer to resolution (FWHM) or mass error? Is it really Da or m/z? It would be less ambiguous to report dimensionless resolving power values and ppm mass errors.

On line 24 of page 6, the authors write that broad native MS peaks were observed for the ACE2 dimer. I'm not sure I would agree as the ions seem very well desolvated and the individual glycosylation and truncation states are clearly distinguishable, even if they are not quite baseline-resolved (compare for example the uninterpretable broad native signal in Figure 3b).

On line 18 of page 8, the authors write that including internal fragments led to a 'combined sequence coverage of 30.9% for the dimeric 171 kDa complex.' Perhaps I missed it, but it would be useful to add the sequence coverage obtained without internal fragments in this sentence. Also, while for a heterodimer a 'combined sequence coverage' of the complex would make sense, for a homodimer it seems equivalent to just writing that a sequence coverage of 30.9% was obtained, unless there is some nuance I missed.

In the Discussion, the authors write that in their experience, directed cleavage occurs when there are no mobile charge carriers available. This has been shown many times in the literature for native or charge-reduced precursors, so the authors could add a few references here.

Line 27 on page 14: 'map'  'maps'

On page 18 (line 35), the authors make it explicit that their method is purely fragment-centric and does not attempt to identify exact proteoforms. This is fine, as a list of identified modifications and potentially their abundances is still very valuable information. However, if that is the case, writing that 'the abundances of different proteoforms were determined' on page 22 (line 36) doesn't seem quite correct, so this should be rephrased to be more precise.

On page 20 (lines 28-33) the authors describe the search for specific variants of previously assigned unmodified ions. If I understand correctly, this is done prior to recalibration, but by adding or subtracting expected mass differences from the measured mass of the unmodified fragments, rather than searching for the theoretical mass of the modified fragments, and this (searching for a specific mass difference rather than a specific mass) is why the error tolerance can be more stringent. I had to reread this a few times though and initially thought that this step happened after recalibration based on the unmodified fragments, so the authors could add a sentence for clarification.

Extended Data Figure 8 is mislabelled as Figure 7.

Reviewer #2 (Remarks to the Author):

Bennet et al. present an interesting and well-designed informatic platform to characterize native protein complexes using tandem mass spectra. Their approach does not require high resolution/accurate intact mass information as a starting point, which differs from mass proteomic tools, but instead uses an iterative approach to assign sequences and modifications to poorly resolved protein complexes using high resolution/accurate mass tandem mass spectra. They first select isotope clusters in tandem MS spectra using the combination of multiple deconvolution algorithms, where correct isotopic envelopes are selected using a machine learning-based supervised voting classifier trained on spectrum-specific data. They then look for sequence tags that could occur from regions of high fragmentation due to structural features or an open search to identify sequence-driven fragmentation. Once protein identifications are assigned, spectra are then re-evaluated to look for the presence of modified ions or unexpected internal fragments, where assignments can be further verified by manual inspection. Their approach to identify these modified or unexpected fragment ions includes several strategies that integrate a multi-notched search and (semi-)continuous scans of the mass offset range. They use this approach to characterize several interesting modifications on protein complexes, highlighting the utility and flexibility of their approach. Critically, this informatics tool will be open-source and freely available. Overall, I see this as a highly interesting and valuable addition to the native top down proteomics informatic landscape. Before I see the manuscript as suitable for publication, however, I have the following questions and comments for the authors to consider.

1. The authors provide a discussion of ion-level false discovery rates, but it seems that *precisION* will always provide some form of identification, even if there is not sufficient information to warrant such assignments. For example, if the wrong protein sequence is assigned in early steps, this will certainly cause major errors when recalibrating to assigned ions, performing multinotched searches, and more. How have the authors approached this challenge? Entrapment style evaluations would be highly informative to see for this software, including searches where the correct protein sequence with improper truncations etc were used and where impossible protein sequences were used to process data. How many fragment ions are incorrectly assigned to impossible outcomes? What do results look like when they should be null? Because *precisION* is designed to provide an answer, knowing what it reports when the answer should be "nothing" is critical to know.

2. Similarly, have the authors checked their assignments with *precisION* using MSn approaches for specific assigned ions, especially those that contain unexpected modifications? For example, if ions represented glycosylated or lipidated species were reselected and fragmented, would they be able to confirm their assignment. For internal fragments, which are very often misassigned with current tools, would MSn provide more evidence that these assignments are valid? I suggest the authors perform these type of confirmatory MSn experiments for every modified/unexpected ion type they report in this manuscript and use that information to report false discovery rate approximations. It is perhaps the internal sequence ions I am most skeptical of, especially since they report 100s of them in each example. The likelihood of misassignment for these ions is high, even as the authors indicate when discussing the use of shared termini filter to reduced the internal fragment FDR from more than 40%

to less than 6.7% (still quite high).

3. Similar to how the isolation widths are provided for each spectrum, knowing how many transients were averaged (methods say up to 1000) for each spectrum would be helpful.

Reviewer #2 (Remarks on code availability):

It is listed as available, but I did not access it at this time.

Reviewer #2 (Remarks on figshare data availability):

This appears to be high quality data in line with claims in the manuscript.

Reviewer #3 (Remarks to the Author):

This paper introduces *precis*ION, a computational framework designed to enhance the analysis of native top-down mass spectrometry data for uncovering hidden post-translational modifications in protein complexes. By employing fragment-level open searches, machine learning-based isotopic envelope filtering, and hierarchical spectral deconvolution, *precis*ION addresses challenges such as low-abundance modifications and sample heterogeneity. The tool was validated on three targets: ACE2 (revealing glycan heterogeneity and truncations), osteopontin (detecting phosphorylation and proteolytic processing), and GAT1 (identifying diverse lipidations), demonstrating improved sequence coverage and integration with cryo-EM data to resolve structural ambiguities. As an open-source platform, *precis*ION aims to advance structural biology and drug discovery by linking PTMs to functional protein states. The work is well executed and an impressive advance for native top-down MS workflows.

Specific comments:

- Page 2: Please re-phrase: “semi-automated analyses of intact protein fragments” to “semi-automated analysis of fragments from intact proteins”. The fragments are not intact species; they are derived from intact proteins
- Page 18: The machine learning model for envelope filtering seems to be spectrum specific. Could this approach overfit individual datasets, limiting its robustness across instruments or protein systems (e.g., membrane vs. soluble proteins)?
- Page 23: Though PTM quantification is described, details on reproducibility (e.g., standard deviations for GAT1 lipid abundances) are sparse. How consistent are these measurements across replicates or experimental conditions?
- General: The system requirements (16 GB RAM) are rather substantial. A brief description of data processing speed would be beneficial to familiarize non-expert users with expected workflow times.
- I applaud the authors for their descriptions of deconvolution processes, and the validation of internal fragments. However, the few groups that include internal ions seem to use a mass tolerance of 2 ppm to reduce the huge number of false positives. This seems even more critical when one is trying to identify/localize PTMs and extra confidence is essential. Please address this. Even a comparison using a mass tolerance of 2 ppm versus 3 ppm for one search would be eye-opening. I know that they are doing other validation steps, like the FDR and using a scrambled sequence, but the use of internal ions opens another potential source of many false hits.
- More examples of isotope distributions of selected fragment ions, the fits compared to theoretical ions, and the scores obtained for those ions (including ones that are retained and ones that are excluded because they fail the cutoff test) should be included. A small collection of these (4-6) could be easily included in a supporting figure.
- I believe the authors miss an opportunity to indicate whether this workflow could be used for standard top-down proteomics (for denatured proteins, not intact protein complexes). Some of the features are geared towards multimeric analysis, but it seems like a lot of the strategy could facilitate top-down analysis of intact proteins.

Reviewer #4 (Remarks to the Author):

A. The authors report a software, *precis*ION, to better define modified proteoforms in native top-down mass spectrometry. Central to the approach is a focus on fragment ions rather than the parent ion.

B. The work is original and significant.

C&D. The authors report a false discovery rate of 5%. I find this unacceptable, it is similar to that obtained in bottom-up experiments. Surely the accuracy of top-down data offers the opportunity to move into the realm where false discovery is vanishingly rare (say 1 in 1000). This would provide a very good reason to perform top-down MS compared to bottom-up. This situation becomes worse when considering internal fragments – the authors report ‘improving FDR from more than 40% to less than 6.7%’. The authors recognize that identifying a series of ions with a common internal terminus is valuable but don’t provide a statistical metric to reinforce this.

Use of a ‘tight mass tolerance (3 ppm; Supplementary Fig. 7).’ is welcomed. Making the most of available accuracy is applaudable and resorting to internal calibration is valued. Supplementary Figure 3 nicely emphasizes performance of latest orbitrap instrumentation.

Switching units of mass error is distracting – pick one and stick with it.

'Peaks corresponding to monoisotopic errors (± 0.0010 Da) and CO loss/a-type ions (± 0.0009 Da) were observed' versus 'mass of palmitate (1 mDa error) and stearate (5 mDa error)', for example.

E. It is understandable why the authors use a human cell line, but deriving any useful physiological meaning should be avoided.

'Thus, it appears that while SPP1 has the potential to be extensively phosphorylated, such modifications may only occur in response to specific stimuli.'

F. One could imagine an overall performance score for the experiment. Data sets where there are too high a proportion of unassigned/internal peaks could be flagged.

G. Key early top-down membrane protein papers are not cited. Collision energies were minimized to completely avoid internal fragments.

H. The body text (results/discussion) is somewhat fuzzy and fails to emphasize the points that are dealt with in more detail in methods/supplemental sections.

Version 2:

Decision Letter:

Our ref: NMETH-A59067B

13th Jun 2025

Dear Carol,

Thank you for submitting your revised manuscript "Uncovering hidden protein modifications with native top-down mass spectrometry" (NMETH-A59067B). It has now been seen by the original referees and their comments are below. The reviewers find that the paper has improved in revision, and therefore we'll be happy in principle to publish it in Nature Methods, pending minor revisions to satisfy the referees' final requests and to comply with our editorial and formatting guidelines.

TRANSPARENT PEER REVIEW

ORCID

Sincerely,
Arunima

Arunima Singh, Ph.D.
Senior Editor
Nature Methods

Reviewer #1 (Remarks to the Author):

With this revised manuscript, the authors have made a sincere effort to address my concerns, including by performing additional experiments and analysis. I believe the precisION software package will have an important impact in native as well as denaturing TDMS.

Reviewer #2 (Remarks to the Author):

The authors have put forth considerable effort to address my comments, and I believe they have improved both their approach and the manuscript. I see this work as now suitable for publication.

Reviewer #2 (Remarks on figshare data availability):

This appears to be high quality data in line with claims in the manuscript.

Reviewer #3 (Remarks to the Author):

The authors have addressed all concerns in a satisfactory manner, and the paper is suitable for publication. The topic is timely, and the reported strategy is compelling. I anticipate that the paper will generate significant interest from the mass spectrometry community.

Reviewer #4 (Remarks to the Author):

- A. The manuscript details an advanced algorithm for assignment of top-down MSMS data that extends the state of the art significantly.
- B. The algorithm is sufficiently advanced compared to previous programs that it can be regarded as original and significant.
- C. The figures (main and supplementary) demonstrate the validity of the approach and the quality of data produced on current advanced instrumentation.
- D. I am happy with use of statistics and treatment of uncertainties.
- E. The conclusions are robust and reliable. I see no reason to doubt the validity of the conclusions.
- F. Though beyond the scope of this manuscript I hope that future studies will use truly native proteins rather than those with sequences modified by mutagenesis and expressed in cell lines.
- G. Appropriate credit was given to previous work in the references.
- H. Clarity and context was excellent. The lucidity and appropriateness of the abstract, introduction and conclusions was very good.

Reviewer #4 (Remarks on code availability):

I do not have credentials to review code.

Version 3:

Decision Letter:

21st Aug 2025

Dear Carol,

I am pleased to inform you that your Article, "Uncovering hidden protein modifications with native top-down mass spectrometry", has now been accepted for publication in Nature Methods. The received and accepted dates will be December 11, 2024 and August 21, 2025. This note is intended to let you know what to expect from us over the next month or so, and to let you know where to address any further questions.

Over the next few weeks, your paper will be copyedited to ensure that it conforms to Nature Methods style. Once your paper is typeset, you will receive an email with a link to choose the appropriate publishing options for your paper and our Author Services team will be in touch regarding any additional information that may be required. It is extremely important that you let us know now whether you will be difficult to contact over the next month. If this is the case, we ask that you send us the contact information (email, phone and fax) of someone who will be able to check the proofs and deal with any last-minute problems.

Authors may need to take specific actions to achieve compliance with funder and institutional open access mandates.

If your research is supported by a funder that requires immediate open access (e.g. according to [Plan S principles](https://www.springernature.com/gp/open-science/plan-s-compliance) or the [NIH public access policy](https://www.springernature.com/gp/open-science/us-federal-agency-compliance)) then you should select the gold OA route, and we will direct you to the compliant route where possible. Because authors warrant under our subscription licensing terms that they haven't committed to licensing any version of their article under a licence inconsistent with the terms of our agreement – including the applicable embargo period – publication under the subscription model isn't suitable for authors whose funders require no embargo.

If you are active on Twitter/X or Bluesky, please e-mail me your and your coauthors' handles so that we may tag you when the paper is published.

Best regards,
Arunima

Arunima Singh, Ph.D.
Senior Editor
Nature Methods

** Visit the Springer Nature Editorial and Publishing website at http://editorial-jobs.springernature.com?utm_source=ejP_NMeth_email&utm_medium=ejP_NMeth_email&utm_campaign=ejp_Nmeth for more information about our career opportunities. If you have any questions please click [here](mailto:editorial.publishing.jobs@springernature.com).

Response to Reviewers' Comments:

Editor's Comments:

Your Article, "Uncovering hidden protein modifications with native top-down mass spectrometry", has now been seen by 4 reviewers. As you will see from their comments below, although the reviewers find your work of considerable potential interest, they have raised a number of concerns. We are interested in the possibility of publishing your paper in Nature Methods, but would like to consider your response to these concerns before we reach a final decision on publication.

We therefore invite you to revise your manuscript to address these concerns. In particular, it would be important to highlight the utility and generalizability of the approach (with new demonstrations, if feasible) in addition to addressing all the other concerns raised by the reviewers.

Thank you for considering our manuscript for publication in Nature Methods. We appreciate the reviewers' thoughtful comments. We have carefully addressed all concerns and included new demonstrations in our revised manuscript.

To better highlight the utility and generalisability of *precis*ION, we now demonstrate the extension of our methods to denatured protein precursors (**Extended Data Fig. 7**), expanding beyond native protein analysis to conventional top-down proteomics workflows. By facilitating more complete interpretation of spectral data, our approach should increase coverage in top-down proteomics and enhance understandings of proteome remodelling in disease. In particular, we believe that the fragment-level open search will enable new methods including top-down glycoproteomics. It will also facilitate the investigation of other labile modifications, such as ADP-ribosylation, and how such modifications 'communicate' with other PTMs. These potential applications are now described in the Discussion section of the revised manuscript, reading "*Our methods also demonstrate significant potential beyond native protein analysis, extending to high-throughput top-down proteomics. The fragment-level open search will enable top-down N-glycoproteomics, uncovering how glycans and other labile modifications communicate with other PTMs to facilitate precise molecular control.*"

Additionally, we demonstrate the power of our methods for dissecting the composition of heterocomplexes (**Extended Data Fig. 1**). Through complete nTDMS analysis of endogenous bovine PDE6 (a heterotetramer), we have shown that *precis*ION can uncover multiple modifications that coexist within a single protein complex, confirming their co-occupancy and role in complex assembly. As indicated by Reviewer #1, this is one of the clearest use cases where nTDMS can add immense value to structural studies.

These additional examples have been included alongside further experiments suggested by the reviewers, discussed in our point-by-point responses. We have also rephrased statements throughout the manuscript to further demonstrate the utility of *precis*ION for the broader research community. Specifically, we highlight how the software package "*offers an intuitive means for interpreting complex protein fragmentation data*" that should democratise detailed analyses of TDMS spectra.

We believe these revisions substantially strengthen our manuscript and demonstrate the broad utility of *precis*ION for advancing integrative structural biology, molecular pathology, and drug development research. We look forward to your feedback on our revised submission.

Reviewers' Comments:

Reviewer #1 (Remarks to the Author):

The authors present a computational method, precisION, to facilitate processing of native top-down mass spectra. They demonstrate their software using three proteins, one of which is a homodimer and the other two are monomeric. In each of the three cases, the samples contain a mix of proteoforms arising due to truncation, glycosylation, phosphorylation, etc. The ability of the precisION software to deal with complex fragmentation spectra is impressive and this tool will be highly useful to the community; however, I have a few comments the authors could consider.

Thank you for your supportive comments. We hope we were able to sufficiently address your concerns and queries.

The authors make the valid point in the introduction that approaches such as deglycosylation and in-solution disassembly sacrifice biological context and insight. Indeed, native (top-down) MS can allow the correlation of proteoform identity to the ability to form noncovalent assemblies. As such, I wonder if the three model systems chosen by the authors really showcase the full capabilities of precisION to provide biological insight, as only one of the models is a complex. Even in that case (a homodimer), it is not clear whether this dataset allows linking of proteoform identity to a tendency to engage in dimer formation. For example, I am curious whether the monomers in Figure 2b exhibit a different glycosylation pattern than the proteoforms incorporated in the dimer.

We agree with the reviewer and recognise that the link between proteoforms and complex formation was not made clear in the original manuscript. The ACE2 N432KO homodimer was specifically selected because of the established effects of modifications on ACE2 dimerisation. We previously used this construct to interrogate the effects of glycosylation (measured using traditional bottom-up glycoproteomics) on ACE2 interactions (10.1021/jacs.3c00291). In this previous study, we demonstrated that *N*-glycans mediate ACE2 self-association. Recognising that this link was not made clear in the original manuscript, we have now defined the importance of glycoforms in ACE2 assembly more explicitly on page 6: “*Here, we selected an isoform of the human angiotensin-converting enzyme 2 (ACE2; Fig. 2a) since it contains six putative N-linked glycosylation sites per subunit and the occupancy of the glycosylation sites influence its dimerization*⁴⁷. **Specifically, distinct glycoforms of ACE2 have been shown to be differentially incorporated into dimers**⁴⁷.”

In this study, we built on our original findings by using precisION to more comprehensively characterise the ACE2 complex. Our analysis uncovered previously undetected C-terminal truncations (not resolved in the original study) and precisely localised *N*-glycans using fragment ions alone. We believe the most significant finding from this study is the ability to interrogate the composition and configuration of *N*-glycoproteins with nTDMS. In future studies, it is our intention that precisION will allow researchers to connect site-specific glycosylation with complex formation propensities, thereby defining meaningful relationships between PTMs and biomolecular interactions.

To address the question regarding differential glycosylation affecting incorporation into dimers, we reanalysed our data comparing the observed glycoform distribution in dimers with that expected assuming equal dimerisation propensity for all glycoforms (**Response Fig. 1**). The observed deviation from the model is consistent with our previous finding that glycan composition affects dimerisation efficiency.

Response Figure 1: Differential incorporation of ACE2 glycoforms into homodimers. Comparison between the observed mole fraction of ACE2 dimers with different total *N*-glycan counts (blue line) and the expected distribution based on a multinomial model (orange line). The multinomial model assumes random pairing of monomeric glycoforms with equal dimerisation propensity for all glycoforms present in the sample. The deviation between observed and predicted distributions indicates that glycan composition influences dimerisation efficiency, with specific glycoforms being preferentially incorporated into dimers.

The authors might also consider including data on a heterocomplex as a demonstration of the use of this software in an experiment in which native top-down MS provides clear added value.

We agree with the reviewer that heterocomplexes would serve as a useful test case for our workflow. Accordingly, we have now included a more thorough analysis of the endogenous heterotetrameric bovine PDE6 complex in **Extended Data Fig.1**. This example was previously summarised in **Fig. 1a**. However, we now recognise that the original representation was overly schematic and did not provide enough details for the reader to appreciate the extensive value of nTDMS in the analysis of multiprotein complexes.

In this new **Extended Data Fig. 1** and associated **Supplementary Note 1** we demonstrate that *precis*ION can successfully identify all three subunits of the heterotetrameric PDE6 complex as the top scoring spectrum matches. We demonstrate good coverage across the length of all three subunits, and show that N-terminal modifications and C-terminal lipidation can be detected using the fragment-level open search. The lack of unmodified forms of these proteins, from an endogenous source, highlights the importance of these modifications in complex maturation and assembly.

Somewhat related to my previous point: As two of the proteins chosen by the authors to demonstrate *precis*ION are monomers, I am curious to know whether the software can also tackle top-down spectra of denatured precursors, possibly with a broad isolation window. It is not obvious to me from the description of the software that this wouldn't be possible (although I suppose fragmentation might be a bit less directed and filtering internal fragments might be more difficult as a result), but as the majority of top-down MS is still carried out in denaturing mode, this seems worth testing to expand the potential user base of *precis*ION. If the authors have tested this, or if it is fundamentally impossible to use this software with denatured precursors, it would be good to note this in the manuscript.

We thank the reviewer for their helpful suggestion. We agree that the algorithms underlying *precis*ION could be readily applied to denaturing TDMS measurements. To demonstrate this versatility, we have now included a new example (**Extended Data Fig. 7**) detailing TDMS analysis of a denatured glycoprotein, avidin.

Using *precis*ION, we identified that avidin harbours heterogeneous glycosylation at one *N*-linked site and is truncated in this preparation. Importantly, we observed the diagnostic HexNAc remainder mass (203 Da) on *b*-type sequence ions, enabling us to localise the *N*-glycan to a specific Asn residue by interpreting fragments alone. We also detected sequence ions retaining larger glycan structures at low HCD acceleration voltages. These insights from *precis*ION enabled us to assign the entire MS¹ spectrum, providing a comprehensive overview of avidin glycoforms. In particular, by examining the fragments retaining larger glycans we were able to support the glycoform assignments made at the MS¹ level by measuring the mass of the *N*-linked glycans with greater precision.

As suggested, fragmentation was less directed than in nTDMS due to the high number of mobile protons present on the unfolded precursor ions. Consequently, we could not assign as many internal fragment ions as we typically observe under high-energy activation of native proteins. This is evident in the fragment map (**Extended Data Fig. 7e**), where relatively few internal fragments are assigned. This suggests that other strategies will be needed for the accurate assignment of internal fragments from denatured proteins.

The reviewer's suggestion has revealed an important potential application of *precis*ION's methods. The fragment-level open search could be readily integrated into high-throughput top-down search engines to enable methods such as top-down *N*-glycoproteomics. Our approach builds upon current informatic methods for database-independent modification discovery (e.g. TopPIC and ProSight Subsequence/DeltaM mode) in top-down proteomics. Our approach is complementary in areas where current methods are limited by (i) their reliance on accurate intact precursor monoisotopic masses (often impractical for large/heterogeneous precursors >30 kDa, [10.1016/j.bbapap.2022.140758](https://doi.org/10.1016/j.bbapap.2022.140758)) and/or (ii) the requirement that modifications remain intact on fragment ions. This additional use case for *precis*ION is now presented in the revised Discussion section, reading: "*Our methods also demonstrate significant potential beyond native protein analysis, extending to high-throughput top-down proteomics. The fragment-level open search will enable top-down N-glycoproteomics, uncovering how glycans and other labile modifications communicate with other PTMs to facilitate precise molecular control.*"

The other methods described in this manuscript are also not limited by the denatured/native state of the precursor. However, some of the scoring functions used throughout the software package are optimised for nTDMS and would be less useful for denatured protein analysis.

Minor points:

Top-down MS of a mixture of unresolved proteoforms, or of a complex consisting of different proteoforms can to an extent be seen as a special case of data-independent analysis and the challenges associated with the chimeric spectra generated in top-down DIA. While the *precis*ION framework seems like a powerful new approach, there is considerable literature about this already, which the authors should at least briefly discuss and cite.

Top-down DIA is a relatively new approach in top-down proteomics that, to our knowledge, was first practically realised at the end of last year. We are aware of two publications describing this method: TopDIA ([10.1021/acs.jproteome.4c00293](https://doi.org/10.1021/acs.jproteome.4c00293)) and Full Window DIA ([10.1021/acs.analchem.4c06471](https://doi.org/10.1021/acs.analchem.4c06471)).

1. Full Window DIA sets out to demultiplex fragment spectra acquired without isolation by computing "elution profiles" for both precursors and fragments, then employing correlation analysis to pair fragments with their respective precursors. The demultiplexed spectra are then processed with the standard TopPIC search engine.
2. TopDIA similarly demultiplexes MS² spectra, but uses 80 Th MS¹ gas-phase fractionation and 4 Th isolation windows (20 MS² scans per cycle) during acquisition. Additionally,

TopDIA employs a more complex three-step workflow to pair precursors with fragments that includes a logistic regression model and more extensive filtering.

It is important to note that both of these methods currently require LC-MS/MS data to demultiplex fragment spectra and are likely to struggle with “kindred” proteoforms which will often co-elute and share many sequence ions (10.1021/acs.jproteome.3c00416), preventing successful demultiplexing. *precis*ION does not rely on LC profiles to demultiplex fragment spectra as this is not possible in the cases of intact heteromeric complexes. Instead, more similar to our approach is TopMPI, a new informatic method (preprint; 10.1101/2025.02.05.636727) that is designed to identify proteoforms from chimeric MS² spectra using an iterative, hierarchical approach that is conceptually similar to *precis*ION’s assignment strategy. Although this method has not yet been demonstrated with DIA TDMS, we believe TopMPI could be easily adapted to address the unique challenges of multiplexed DIA data and overcome some of the aforementioned limitations. We acknowledge the conceptual similarity between some of these new DIA TDMS methods and *precis*ION’s approaches and have now included the following statement in the revised manuscript on page 21 line 10: “*To prevent repeated matching, each assigned fragment ion is removed from the list of envelopes in a similar manner to TopMPI⁸⁷, a recently reported approach for the analysis of chimeric intact protein fragmentation spectra that may facilitate top-down data-independent acquisition MS.⁸⁸*” This includes citations to both TopMPI and TopDIA.

In the caption for Figure 1, the authors write that mass resolution for intact complexes is fairly low (10-500 Da). Do they really mean Da, or m/z? Similarly, they write that fragment ions are ‘measured with high mass resolution and accuracy (typically < 0.02 Da)’. Does that value refer to resolution (FWHM) or mass error? Is it really Da or m/z? It would be less ambiguous to report dimensionless resolving power values and ppm mass errors.

We appreciate the reviewer's concern about precision in describing mass spectrometry measurements. However, we deliberately chose not to report dimensionless resolving powers or relative ppm mass errors in this section, as these parameters would be challenging for the general readership of *Nature Methods*. Those new to the field may not immediately understand what “3 ppm mass accuracy” or “R = 240,000 @ m/z 200” means practically in terms of confidently identifying PTMs and discriminating between similar modifications.

Instead, we have updated the manuscript on page 3 to read: “*Intact protein complexes, exhibiting peak widths of 10–1,000 Da FWHM in deconvolved spectra, are selected and activated in the gas phase, yielding fragment ions measured with both: (1) sufficient resolving power to distinguish between similar protein modifications, and (2) high mass accuracy (errors typically less than 20 mDa) to confidently identify these modifications.*” This statement is also supported by detailed analysis of the PDE6 spectra in **Supplementary Note 1**.

This change maintains the absolute units (Da) that directly relate to modification mass shifts. This approach provides both technical accuracy and accessibility for the broader scientific audience.

On line 24 of page 6, the authors write that broad native MS peaks were observed for the ACE2 dimer. I’m not sure I would agree as the ions seem very well desolvated and the individual glycosylation and truncation states are clearly distinguishable, even if they are not quite baseline-resolved (compare for example the uninterpretable broad native signal in Figure 3b).

We agree that the original description incorrectly referred to the individual peaks as ‘broad’ and have updated the manuscript to read: “*Even when produced in HEK293 GNTI^{-/-} cells, in which all N-glycans have a fixed composition (Man₅GlcNAc₂), complex native MS peak patterns were observed for ACE2 dimers, suggesting heterogeneity (Fig. 2b).*”

On line 18 of page 8, the authors write that including internal fragments led to a 'combined sequence coverage of 30.9% for the dimeric 171 kDa complex.' Perhaps I missed it, but it would be useful to add the sequence coverage obtained without internal fragments in this sentence. Also, while for a heterodimer a 'combined sequence coverage' of the complex would make sense, for a homodimer it seems equivalent to just writing that a sequence coverage of 30.9% was obtained, unless there is some nuance I missed.

We originally used the term 'combined' to indicate the sequence coverage obtained with terminal and internal fragments across all collision energies. However, we now recognise that this term could be easily misunderstood and have updated the phrase on page 8 to read: "*As a result, we significantly increased the fraction of the spectrum that could be readily interpreted, particularly at high HCD energies, facilitating the assignment of an additional 632 fragment ions and a final sequence coverage of 30.9% (an increase from 10.6% when not considering internal fragments) for the dimeric 171 kDa complex (Fig. 2d, Supplementary Fig. 4).*"

The revised sentence also indicates the sequence coverage obtained when not considering internal fragments.

In the Discussion, the authors write that in their experience, directed cleavage occurs when there are no mobile charge carriers available. This has been shown many times in the literature for native or charge-reduced precursors, so the authors could add a few references here.

We agree with the reviewer's suggestion and have now cited several papers (refs 71-73) describing the mechanisms of directed fragmentation of peptides and proteins in the revised Discussion on page 14.

Line 27 on page 14: 'map'  'maps'

We have made this change.

On page 18 (line 35), the authors make it explicit that their method is purely fragment-centric and does not attempt to identify exact proteoforms. This is fine, as a list of identified modifications and potentially their abundances is still very valuable information. However, if that is the case, writing that 'the abundances of different proteoforms were determined' on page 22 (line 36) doesn't seem quite correct, so this should be rephrased to be more precise.

We have now updated the respective statement on page 23 to "*We quantified the abundance of different modifications from MS² spectra using precisiON's Proteoform analysis module.*"

On page 20 (lines 28-33) the authors describe the search for specific variants of previously assigned unmodified ions. If I understand correctly, this is done prior to recalibration, but by adding or subtracting expected mass differences from the measured mass of the unmodified fragments, rather than searching for the theoretical mass of the modified fragments, and this (searching for a specific mass difference rather than a specific mass) is why the error tolerance can be more stringent. I had to reread this a few times though and initially thought that this step happened after recalibration based on the unmodified fragments, so the authors could add a sentence for clarification.

The reviewer's understanding of our approach is correct. We recognise that this was not sufficiently clear in the original description and have updated the text accordingly on page 21. The revised manuscript reads: "*After assigning the first set of unmodified ions, precisiON considers potential variants of these ions, specifically dehydrated, deamidated, and sodiated forms. In this phase, precisiON employs finer matching tolerances (e.g., 3 ppm), **which are permitted by searching for ions with defined mass differences from the assigned ions, correcting for systematic mass errors.** This cycle iterates multiple times until no further products can be assigned, ensuring that the*

major side products of fragmentation upon vibrational activation are assigned without unnecessarily widening the search space.”

Extended Data Figure 8 is mislabelled as Figure 7.

We thank the reviewer for bringing this to our attention. The incorrectly labelled Extended Data Figure has now been correctly labelled.

Reviewer #2 (Remarks to the Author):

Bennet et al. present an interesting and well-designed informatic platform to characterize native protein complexes using tandem mass spectra. Their approach does not require high resolution/accurate intact mass information as a starting point, which differs from mass proteomic tools, but instead uses an iterative approach to assign sequences and modifications to poorly resolved protein complexes using high resolution/accurate mass tandem mass spectra. They first select isotope clusters in tandem MS spectra using the combination of multiple deconvolution algorithms, where correct isotopic envelopes are selected using a machine learning-based supervised voting classifier trained on spectrum-specific data. They then look for sequence tags that could occur from regions of high fragmentation due to structural features or an open search to identify sequence-driven fragmentation. Once protein identifications are assigned, spectra are then re-evaluated to look for the presence of modified ions or unexpected internal fragments, where assignments can be further verified by manual inspection. Their approach to identify these modified or unexpected fragment ions includes several strategies that integrate a multi-notched search and (semi-)continuous scans of the mass offset range. They use this approach to characterize several interesting modifications on protein complexes, highlighting the utility and flexibility of their approach. Critically, this informatics tool will be open-source and freely available. Overall, I see this as a highly interesting and valuable addition to the native top down proteomics informatic landscape. Before I see the manuscript as suitable for publication, however, I have the following questions and comments for the authors to consider.

We thank the reviewer for recognising the value our approach adds to the nTDMS methods landscape. We have made several alterations to the manuscript and software to address their questions and comments which are detailed below.

1. The authors provide a discussion of ion-level false discovery rates, but it seems that *precis*ION will always provide some form of identification, even if there is not sufficient information to warrant such assignments. For example, if the wrong protein sequence is assigned in early steps, this will certainly cause major errors when recalibrating to assigned ions, performing multinotched searches, and more. How have the authors approached this challenge? Entrapment style evaluations would be highly informative to see for this software, including searches where the correct protein sequence with improper truncations etc were used and where impossible protein sequences were used to process data. How many fragment ions are incorrectly assigned to impossible outcomes? What do results look like when they should be null? Because *precis*ION is designed to provide an answer, knowing what it reports when the answer should be “nothing” is critical to know.

As the reviewer notes, *precis*ION is designed to always attempt an identification from MS² spectra, even in challenging cases. In the original version of the software, users could assess the quality of putative matches using a range of quantitative metrics (e.g., nFPS, sequence coverage, % signal assigned, #*b*-/*y*-/*l*-type ions) alongside visual tools (e.g., annotated spectra, fragment maps, calibration curves, fragmentation propensity heatmaps, 3D structure). However, we recognise that less experienced users may find it difficult to interpret ambiguous results from these metrics alone.

To mitigate this, we have now implemented a set of automated warning flags that are triggered by results that appear consistent with common modes of failure. These include:

1. **Low proportion of signal assigned:** “Less than a third of the deconvoluted signal has been assigned. A low percentage of assigned signal can indicate that the assigned sequence(s) are incorrect. Care should be taken when interpreting the resulting data. However, it may also indicate the presence of other biomolecules and/or their fragments in the spectrum that have been left unassigned. To check this, examine the spectrum using Fragment Assignment -> Show Spectrum to identify prominent unassigned peaks. If these peaks cannot be reliably assigned to protein fragments (i.e., they do not correspond to other unassigned proteins or modified sequence ions), they may be identified using MS3 measurements along with de novo peptide sequencing or metabolite annotation tools such as SIRIUS.”
2. **Number of internal fragments is greater than twice the number of terminal fragments:** “A very large number of internal fragments have been assigned. This may indicate the collision energy is too high to observe labile modifications.”
3. **Number of b-type ions is less than 5 (or the same for y-type ions):** “Less than 5 b-type ions have been assigned. This may be due to: 1) Incorrect N-terminal sequence, modification state, or N-terminus. 2) An uneven charge distribution. To check the distribution of charges along the protein, use the Fragment Analysis -> Intensity Histogram plot.”
4. **Number of b-type ions is less than three times the count of y-type ions (and visa-versa):** “b-type ions significantly outnumber y-type ions. This may be due to: 1) Incorrect C-terminal sequence, modification state, or C-terminus. 2) An uneven charge distribution. To check the distribution of charges along the protein, use the Fragment Analysis -> Intensity Histogram plot.”

In response to the reviewer’s suggestion, to rigorously evaluate the updated software’s behaviour in the presence of incorrect inputs, we conducted an entrapment-style study, detailed in **Supplementary Figure 19** and **Supplementary Note 2**. These analyses used MS/MS data from native bovine carbonic anhydrase II and explored how precisiON responds under different error scenarios:

1. **Random shuffled sequence:** A shuffled bCA2 sequence was used and the full protocol was followed without human intervention (e.g., the termini that resulted in the greatest number of matches were used, despite their biological implausibility). After the assignment of unmodified sequence ions, no significant mass offsets were observed in the fragment-level open search. Warning flags 1 and 2 were printed.
2. **Incorrect database:** Searching against the human proteome returned AASD1_HUMAN as the top match by nFPS, but this score fell below precisiON’s confidence threshold (“green” nFPS > 5), and the assignment was rejected.
3. **Incorrect termini:** When one or both termini were manually fixed to incorrect positions, warning flags were consistently triggered (low signal assignment, few or imbalanced terminal ions), clearly suggesting a misassignment.
4. **Incorrect variant:** When a known variant of carbonic anhydrase II was used (R57Q), the fragment-level open search returned a significant delta mass (+90.9566 Da) on b-type ions. This mass shift was attributable to the Q>R mutation and Zn binding, demonstrating that precisiON was able to guide the user toward the correct variant, despite the presence of a combined, complex modification. While such a case may be difficult to resolve blindly, the open search made it evident that additional modifications or mutations needed to be considered.

In all cases tested, precisiON did not produce misleading high-confidence identifications. Instead, it raised warning flags that guide the user toward appropriate scrutiny. We believe these additions provide users with sufficient information to identify when results are unsupported, and when the correct conclusion may be that “no valid assignment” can be made.

2. Similarly, have the authors checked their assignments with *precis*ION using MSn approaches for specific assigned ions, especially those that contain unexpected modifications? For example, if ions represented glycosylated or lipidated species were reselected and fragmented, would they be able to confirm their assignment. For internal fragments, which are very often misassigned with current tools, would MSn provide more evidence that these assignments are valid? I suggest the authors perform these type of confirmatory MSn experiments for every modified/unexpected ion type they report in this manuscript and use that information to report false discovery rate approximations. It is perhaps the internal sequence ions I am most skeptical of, especially since they report 100s of them in each example. The likelihood of misassignment for these ions is high, even as the authors indicate when discussing the use of shared termini filter to reduced the internal fragment FDR from more than 40% to less than 6.7% (still quite high).

We thank the reviewer for their insightful suggestion. We strongly agree that experimental confirmation of unexpected fragment ions will help to establish the reliability of *precis*ION's novel fragment-level open search capabilities.

While comprehensive MS³ validation of every reported modified ion was not feasible due to signal constraints, we have substantially expanded our experimental validation in the revised manuscript:

1. We now include MS³ data for lipidated PDE6 sequence ions (**Extended Data Fig. 1**), which supports our assignment of these modified γ -type ions.
2. To strengthen our glycosylation assignments, we analysed an additional glycoprotein (avidin) that exhibits the identical diagnostic 203 Da remainder mass on b -type ions as observed with ACE2 (**Extended Data Fig. 7**). This consistent pattern across independent proteins provides cross-validation for these unexpected ion types.
3. For the lipidated GAT1 γ -type ions, we observe a characteristic palmitoylated/oleoylated/stearoylated envelope pattern consistently across multiple sequence ions (**Supplementary Fig. 15**). This distinctive modification signature appears at precisely the expected mass intervals and relative abundances, providing robust evidence for our assignments without requiring additional MS³ experiments.

Regarding your specific concerns about internal fragments, we acknowledge their higher potential for misassignment and have addressed this through additional targeted experiments:

1. We acquired a new spectrum by isolating native SPP1 ions and subjecting them to HCD with a high 150 V acceleration potential specifically optimised to generate abundant internal fragments (**Supplementary Fig. 17**).
2. Analysis of the fragmentation sites reveals that these internal fragments follow expected cleavage patterns consistent with established fragmentation mechanisms in nTDMS (**Supplementary Fig. 17**).
3. Most critically, we performed extensive MS³ validation by acquiring spectra for the 20 most abundant internal fragments that could be cleanly isolated (**Supplementary Fig. 20**). Using *de novo* sequencing and spectral alignment approaches comparable to established methods (10.1021/acs.analchem.0c04670), we obtained experimental support for 19 of 20 assignments (95% validation rate). The single unconfirmed case produced insufficient fragments for conclusive interpretation, with no alternative assignment emerging from the data.

These data provide strong supporting evidence for our internal fragment assignments and demonstrate the reliability of *precis*ION's approaches, even for these challenging fragment types.

Despite the ability of MSⁿ measurements to confirm fragment assignments, several technical challenges limit the practical application of these approaches for comprehensive validation in large-scale nTDMS analyses:

1. Fragment ion isolation: The dense array of sequence ions in MS² spectra can prevent the clean isolation of individual sequence ions using linear quadrupole ion traps, particularly for low-abundance species (**Supplementary Fig. 17**)
2. Practical constraints: MS³ measurements can take a prohibitively long time, especially when accounting for necessary fragment-by-fragment collision energy optimisation and the inherently low ion abundances in these experiments. For example, for a soluble protein with high signal intensity, we acquire MS³ spectra at four collision energies for each fragment, each of which takes >1 min.
3. Fragmentation limitations: Many low-charge sequence ions that lack mobile protons produce uninformative secondary fragmentation patterns, hindering MS/MS sequencing approaches.

In comparison, *precis*ION offers a more general informatic solution that affords robust, statistically-backed assignments of unexpected ions without the requirement for additional experimental steps. *precis*ION's approach also ensures the method is compatible with a greater range of instrumentation that cannot perform MSⁿ measurements (e.g., Q–Orbitrap or Q–FT-ICR).

3. Similar to how the isolation widths are provided for each spectrum, knowing how many transients were averaged (methods say up to 1000) for each spectrum would be helpful.

We recognise that others attempting to replicate our results, or acquire their own high-quality data, may appreciate extensive descriptions of spectrum-specific instrument parameters. However, the number of transients averaged is unfortunately not stored in Thermo RAW data files, and we did not systematically note this information during acquisition.

We did not record this information because, rather than using a fixed number of transients, our acquisition approach relies on real-time assessment of spectral quality. We monitor the evolving averaged spectrum during acquisition and save spectra once a sufficient signal-to-noise ratio ($S/N \propto \sqrt{n}$) is achieved to resolve low-abundance ion envelopes. This adaptive approach optimises acquisition time while ensuring data quality, but results in sample-dependent variation in the number of transients averaged. To provide practical guidance for researchers seeking to replicate our work, we have updated our methods section on page 17 with relevant details: *“For each spectrum, up to 1,000 transients were averaged to **enhance** the signal-to-noise ratio. **This averaging was implemented through the Tune instrument control software by activating the averaging function (Σ symbol) and setting “Scans to average” to 999. Averaged spectra were saved as .raw files once the signal-to-noise ratio was sufficient to resolve low-abundance envelopes, typically when noise levels decreased below 0.5% relative intensity.”***

We believe this description, combined with the other parameters we have reported and the raw data files, offer sufficient detail for other researchers to replicate our methods.

Reviewer #2 (Remarks on code availability):

It is listed as available, but I did not access it at this time.

Reviewer #2 (Remarks on figshare data availability):

This appears to be high quality data in line with claims in the manuscript.

Reviewer #3 (Remarks to the Author):

This paper introduces *precis*ION, a computational framework designed to enhance the analysis of native top-down mass spectrometry data for uncovering hidden post-translational modifications in protein complexes. By employing fragment-level open searches, machine learning-based isotopic

envelope filtering, and hierarchical spectral deconvolution, precisION addresses challenges such as low-abundance modifications and sample heterogeneity. The tool was validated on three targets: ACE2 (revealing glycan heterogeneity and truncations), osteopontin (detecting phosphorylation and proteolytic processing), and GAT1 (identifying diverse lipidations), demonstrating improved sequence coverage and integration with cryo-EM data to resolve structural ambiguities. As an open-source platform, precisION aims to advance structural biology and drug discovery by linking PTMs to functional protein states. The work is well executed and an impressive advance for native top-down MS workflows.

We thank the reviewer for their supportive comments and for recognising our approach as an advancement in native top-down MS. We have addressed their comments and detailed the changes below.

Specific comments:

- Page 2: Please re-phrase: “semi-automated analyses of intact protein fragments” to “semi-automated analysis of fragments from intact proteins”. The fragments are not intact species; they are derived from intact proteins

We thank the reviewer for their careful examination of the manuscript. We have now made this change.

- Page 18: The machine learning model for envelope filtering seems to be spectrum specific. Could this approach overfit individual datasets, limiting its robustness across instruments or protein systems (e.g., membrane vs. soluble proteins)?

The spectrum-specific nature of our machine learning approach is a deliberate design choice rather than a limitation.

In traditional machine learning applications, overfitting occurs when a model performs well on training data but poorly on unseen data (e.g. other spectra). However, our approach is fundamentally different. We intentionally train a classifier on a subset of manually evaluated envelopes from a specific spectrum to subsequently filter the full complement of envelopes within that same spectrum. The goal is not to create a generalisable model that can be applied across all possible spectra, but rather to optimise envelope classification for an individual dataset (spectrum).

True overfitting in this context would occur if the classifier only performed well on the training subset but failed on the unseen envelopes within the same spectrum. To mitigate this risk, we implement several safeguards:

1. Training is performed on a randomly selected, representative subset of envelopes to ensure the variance of the training set mirrors that of the entire dataset.
2. We use a 75:25 train/test split and provide users with a confusion matrix to quantitatively assess classifier performance
3. Our voting scheme prioritises minimising false negatives, reducing the risk of discarding genuine envelopes even when classification is imperfect

The rationale behind our spectrum-specific approach is that native top-down mass spectra from different protein systems or instruments exhibit dramatically different characteristics (signal-to-noise ratios, peak shapes, envelope distributions, charge states, etc.). A universal model would likely underperform across this diversity. By creating tailored classifiers, we achieve superior envelope classification for each individual dataset, similar to how automated particle picking algorithms in cryoEM are typically trained on specific sets of micrographs to boost performance.

While we agree that a general classifier would be valuable for the community, developing such a model would require extensive training data spanning diverse mass analysers, protein systems, and

spectral qualities. This data is not yet available. Moreover any successful general approach would still need to incorporate some form of initial (potentially implicit) stratification based on instrument parameters or spectral characteristics before considering individual envelopes. precisION's current approach enables high-quality envelope classification while also facilitating the collection of comprehensively annotated spectra that could support future development of more general classifiers.

Despite our intentional design choice, during the development of precisION, we also considered the generality of our classifiers. Briefly, we independently trained three distinct classifiers using three spectra acquired on the same instrument but from different protein systems. We then used each of these classifiers to filter envelopes from each spectrum and compared the envelopes identified in each case (such 'cross-application' of pre-trained classifiers is possible in precisION but not recommended). The resulting Venn diagrams depicting the overlap of each filtered set of envelopes (**Response Fig. 2**) indicate substantial agreement across classifiers, suggesting each classifier recognised similar features that may be fundamental 'true' envelope characteristics. However, there were still noticeable differences between each classifier's selection – we believed these differences were significant enough to warrant spectrum-specific classification. Please note that we did not include this figure in the manuscript as we wish to encourage the use of spectrum-specific classification by precisION's users.

Response Figure 2: Generality of spectrum-specific envelope classifiers. We designed a scheme to test how effectively the spectrum-specific classifiers generated by precisION performed across different biological systems. To do this we: 1) trained three spectrum-specific classifiers using three independent spectra, then 2) used each classifier to filter putative envelopes from each of the three spectra. We compared the overlap between the three sets of envelopes identified for each spectrum, finding substantial overlap but still some significant differences. We believe these differences arise from unique spectral characteristics that are not replicated across the different biological systems.

- Page 23: Though PTM quantification is described, details on reproducibility (e.g., standard deviations for GAT1 lipid abundances) are sparse. How consistent are these measurements across replicates or experimental conditions?

We thank the reviewer for raising this important point. All quantitative bar charts and scatter plots in the manuscript include error bars, which reflect the standard deviation of the measured values ($n \geq 3$). Individual data points are also displayed where possible. These error bars capture the technical variability across replicate spectra and/or emitters, thereby providing a measure of reproducibility across experimental runs.

We agree that the original manuscript provided limited information regarding biological variability. However, the proteins in this study were expressed recombinantly in HEK293 cells under tightly controlled conditions so biological variability is expected to be minimal. To confirm this in the context of GAT1 lipid abundances, we analysed native mass spectra from six independently prepared GAT1 samples, each purified from a separate batch of cells grown and processed on different days. The resulting lipid abundance profiles showed limited batch-to-batch variability, supporting the robustness and reproducibility of our measurements across biological replicates. These additional data are now included in **Supplementary Fig. 16**, as noted in the revised manuscript.

- General: The system requirements (16 GB RAM) are rather substantial. A brief description of data processing speed would be beneficial to familiarize non-expert users with expected workflow times.

precisION was developed and extensively tested on a variety of Windows 10 systems with 16+ GB RAM. While 16 GB RAM is standard in most modern research computing environments, we have now verified that precisION also operates stably on systems with 8 GB RAM and have updated the manuscript accordingly. Our testing revealed that the most memory-intensive task is the open search for protein chain identification, which becomes significant only when searching large databases (e.g. the entire human proteome).

We have also now included a list of anticipated processing times for an example dataset in precisION's README file. This file will serve as the 'home page' of the software on GitHub ensuring its visibility to end users. We believe the processing times are reasonable: rapid deconvolution takes less than a minute, optional extensive deconvolution requires approximately 15 minutes (and can be batched), envelope classification requires about 1.5 minutes for classifier training, and the database search used to identify proteins takes only 15-25 seconds (or 10 s after first-pass initial database generation). The fragment-level open search with a mass offset range of -100 to 500 Da should take approx. 25 s for the example dataset. These benchmarks make precisION suitable for most routine offline nTMS analyses, where acquisition times are typically > 10 minutes. Further software automation will be required if the methods are to be adapted for online, high-throughput analyses – we believe this would be most readily achieved by integrating algorithms into established high-throughput search engines.

- I applaud the authors for their descriptions of deconvolution processes, and the validation of internal fragments. However, the few groups that include internal ions seem to use a mass tolerance of 2 ppm to reduce the huge number of false positives. This seems even more critical when one is trying to identify/localize PTMs and extra confidence is essential. Please address this. Even a comparison using a mass tolerance of 2 ppm versus 3 ppm for one search would be eye-opening. I know that they are doing other validation steps, like the FDR and using a scrambled sequence, but the use of internal ions opens another potential source of many false hits.

We thank the reviewer for their suggestion regarding the effects of mass tolerance on internal fragment assignment. We agree that internal fragment ions could potentially result in many false assignments and have now explored the effect of mass tolerance on the assignment of these ions and the corresponding ion-level false discovery rates.

We chose to use 3 ppm as the 'fine' mass tolerance in this paper based on empirical analyses of high resolution Orbitrap data. When the mass tolerance used for assignment was less than 3 ppm

(e.g., 2 ppm), we often observed two significant peaks on either side of 0.00 Da in the fragment-level open search corresponding to unassigned unmodified *b*-*y*-type ions with mass errors greater than 2 ppm. Increasing the mass tolerance to 3 ppm removed these peaks from the mass offset scan, affording a set of assigned ions with normally-distributed mass errors (**Supplementary Fig. 3**).

To address your concern about internal fragment false positives, we investigated whether tightening the mass tolerance from 3 ppm to 2 ppm would further improve discrimination between true and false internal fragments. Using the ACE2 HCD 180 V dataset (which contains numerous internal fragments due to the high acceleration voltage), we found that without the 'shared termini' filter, the ion-level FDR remained >45% regardless of whether we applied a 3 ppm or 2 ppm mass tolerance (**Supplementary Fig. 6**). While the tighter tolerance did reduce the total number of assigned ions by ~25%, it had minimal impact on the FDR itself. In contrast, applying our 'shared termini' filter with an E-value threshold of 0.01 (precisION's standard settings) reduced the FDR dramatically from 51.7% to 5.9% – an 8.7-fold improvement. Combining the shared termini filter with the tighter 2 ppm tolerance showed little additional benefit in terms of significantly reducing the ion-level FDR.

In conclusion, our data demonstrates that precisION's 'shared termini' filtering approach is far more effective at discriminating true from false internal fragments than simply tightening the mass tolerance. This data (**Supplementary Fig. 6**) validates our approach of using the shared termini filter with a 3 ppm tolerance as an optimal balance between coverage and accuracy for internal fragment assignment.

- More examples of isotope distributions of selected fragment ions, the fits compared to theoretical ions, and the scores obtained for those ions (including ones that are retained and ones that are excluded because they fail the cutoff test) should be included. A small collection of these (4-6) could be easily included in a supporting figure.

In addition to the >10 isotope distributions of assigned ions depicted throughout the manuscript's figures; we have now added a new dedicated figure (**Supplementary Fig. 18**) which displays example envelopes with distinct fit scores to help the reader appreciate the meaning of this value. Please note that there is no defined fit score cut-off applied at any point in precisION's workflow. Rather, during envelope classification most of the low score envelopes will be filtered out by the ML models (see **Extended Data Fig. 3**) and during assignment, any envelope with a fit score less than a user defined threshold (default 0.75) is manually evaluated. In nearly all cases, this results in ~90% of the assigned envelopes exhibiting a fit score greater than 0.7.

- I believe the authors miss an opportunity to indicate whether this workflow could be used for standard top-down proteomics (for denatured proteins, not intact protein complexes). Some of the features are geared towards multimeric analysis, but it seems like a lot of the strategy could facilitate top-down analysis of intact proteins.

Thank you for highlighting this important extension of our work. As discussed in detail in our response to Reviewer #1 on pages 3-4 of our responses, we agree fully agree that the underlying algorithms are readily applicable to conventional top-down proteomics workflows. To demonstrate this versatility, we have now included a new example in the revised manuscript (**Extended Data Fig. 7**) that showcases precisION's performance in analysing denatured avidin, a glycoprotein with a single *N*-glycan. Our results reveal that precisION's fragment-level open search effectively detects modified sequence ions even under denaturing conditions, enabling both determination of *N*-glycan composition and site-specific localisation from standard top-down mass spectra. We have discussed the implications of these methods for standard top-down proteomics in the Discussion of the revised manuscript (page 14) which now reads: "*Our methods also demonstrate significant potential beyond native protein analysis, extending to high-throughput top-down proteomics. The fragment-level open*

search will enable top-down N-glycoproteomics, uncovering how glycans and other labile modifications communicate with other PTMs to facilitate precise molecular control."

Reviewer #4 (Remarks to the Author):

A. The authors report a software, precisION, to better define modified proteoforms in native top-down mass spectrometry. Central to the approach is a focus on fragment ions rather than the parent ion.

B. The work is original and significant.

We thank the reviewer for recognising the originality and significance of our study.

C&D. The authors report a false discovery rate of 5%. I find this unacceptable, it is similar to that obtained in bottom-up experiments. Surely the accuracy of top-down data offers the opportunity to move into the realm where false discovery is vanishingly rare (say 1 in 1000). This would provide a very good reason to perform top-down MS compared to bottom-up. This situation becomes worse when considering internal fragments – the authors report 'improving FDR from more than 40% to less than 6.7%'. The authors recognize that identifying a series of ions with a common internal terminus is valuable but don't provide a statistical metric to reinforce this.

We agree that top-down MS can enable a lower protein-level false discovery rate vs. bottom-up approaches due to the absence of protein inference and the greater amount of information contained within the data. However, we'd like to clarify some important distinctions regarding our reported metrics.

*The ~5% FDR we report refers specifically to **ion-level** false discoveries (individual fragment assignments), not the modification/protein-level FDR. This distinction is crucial. While individual ions might occasionally be misassigned, identifying proteins and their modifications in TDMS requires **consistent patterns to be detected across multiple sequence ions**. Consequently, we anticipate the modification- and protein-level FDR to be substantially lower than the ion-level FDR, approaching "vanishingly rare."*

Specifically, to minimise the false identification of protein modifications and internal fragments in this manuscript, we employ a strict E-value threshold of 0.01 (corresponding to a relatively large number of observations of a particular modification) in our fragment-level open search. This means that we would expect to observe a random match of that significance approximately once in 100 searches (much like a p-value with some small differences). This stringent threshold dramatically reduces the likelihood of false modification discovery, even if some individual ions might be falsely assigned.

*In the context of internal fragments, an E-value threshold is applied to identify sets of internal fragments that share a common terminal fragmentation site. Accordingly, as with modifications, the likelihood of identifying a false **set** of internal fragments is very low due to the strict threshold used. By only assigning ions that fall into one of these high-confidence sets, we can significantly reduce the internal fragment FDR from ~40% in the best case to 6.7% in the worst case. We believe this change is of high practical significance, as it enables proteins to be much more confidently sequenced using internal fragments. We have also now included new empirical analyses to demonstrate the value of this approach for internal fragment assignment (**Supplementary Fig. 6**) and hope the ~10-fold decrease in ion-level FDR provides the necessary statistical evidence to support the analytical value of our method.*

We also compared our reported ion-level false discovery rates to other established approaches to confirm our view that a ~5% ion-level FDR is typical in nTDMS analyses. Specifically, we analysed ion-level false discovery rates calculated by an established software package, TDValidator, for

seven of the spectra evaluated in this paper (**Response Fig. 3**). TDValidator directly fits theoretical sequence ions to raw spectra. In our analysis, we only considered standard *b*-/*y*-type ions (i.e., no internals, or varied modifications) and used the software's standard matching thresholds. At 3 ppm (the mass threshold used in our manuscript); we observed an average ion-level FDR of 2.7%. At 5 ppm (the mass threshold used by many other research groups); the ion-level FDR increased to 4.3%. Considering that *precis*ION expands its outlook well beyond standard *b*-/*y*-type ions to include multiple modifications and internal fragments, we feel that the extensive increase in fragment coverage warrants the very slight increase in ion-level FDR.

Ion-level FDR with single proteoform, standard *b*-/*y*-type ions only
Calculated with TD Validator (shuffled sequence)

Response Figure 3: Ion-level false discovery rates in nTDMS. Bar plot showing ion-level false discovery rates for seven nTDMS spectra analysed in this manuscript. FDRs were calculated using the “shuffled sequence” option in TDValidator with mass tolerances of either 3 or 5 ppm after recalibration. Error bars indicate the standard deviation of the replicate measurements.

We appreciate your emphasis on achieving the lowest possible FDR, which aligns with our goals for *precis*ION. The combination of pattern recognition across multiple ions and stringent statistical thresholds allows us to leverage the unique advantages of top-down MS while maintaining rigorous standards for discovery.

Use of a ‘tight mass tolerance (3 ppm; Supplementary Fig. 7).’ is welcomed. Making the most of available accuracy is applaudable and resorting to internal calibration is valued. Supplementary Figure 3 nicely emphasizes performance of latest orbitrap instrumentation.

We thank the reviewer for this supportive comment. We hope further instrument developments will offer improved mass accuracy. This will enable highly sensitive identification of protein modifications using the fragment-level open search.

Switching units of mass error is distracting – pick one and stick with it.

‘Peaks corresponding to monoisotopic errors (± 0.0010 Da) and CO loss/a-type ions (± 0.0009 Da) were observed’ versus ‘mass of palmitate (1 mDa error) and stearate (5 mDa error)’, for example.

We recognise this inconsistency could prove distracting. All mass errors in the revised manuscript are now expressed in terms of mDa.

E. It is understandable why the authors use a human cell line, but deriving any useful physiological meaning should be avoided.

‘Thus, it appears that while SPP1 has the potential to be extensively phosphorylated, such modifications may only occur in response to specific stimuli.’

We recognise that our data may not reflect PTM regulation under physiological conditions. We have now removed this statement from the revised manuscript.

F. One could imagine an overall performance score for the experiment. Data sets where there are too high a proportion of unassigned/internal peaks could be flagged.

While a single quantitative score to describe experimental success would be valuable for both users and other researchers, we believe that defining a generalizable metric is challenging. The diversity of experimental goals in nTDMS, from isoform identification to structural analysis, means no single number could reliably capture success across all contexts.

Instead, we have implemented a series of qualitative flags (informed by common modes of failure) within the software that provide immediate, interpretable feedback for users, especially those less experienced with spectral interpretation. These are displayed when the user requests a "Spectrum Summary." Examples include:

1. **Low proportion of signal assigned:** *"Less than a third of the deconvoluted signal has been assigned. A low percentage of assigned signal can indicate that the assigned sequence(s) are incorrect. Care should be taken when interpreting the resulting data. However, it may also indicate the presence of other biomolecules and/or their fragments in the spectrum that have been left unassigned. To check this, examine the spectrum using Fragment Assignment -> Show Spectrum to identify prominent unassigned peaks. If these peaks cannot be reliably assigned to protein fragments (i.e., they do not correspond to other unassigned proteins or modified sequence ions), they may be identified using MS3 measurements along with de novo peptide sequencing or metabolite annotation tools such as SIRIUS."*
2. **Number of internal fragments is greater than twice the number of terminal fragments:** *"A very large number of internal fragments have been assigned. This may indicate the collision energy is too high to observe labile modifications."*
3. **Number of b-type ions is less than 5 (or the same for y-type ions):** *"Less than 5 b-type ions have been assigned. This may be due to: 1) Incorrect N-terminal sequence, modification state, or N-terminus. 2) An uneven charge distribution. To check the distribution of charges along the protein, use the Fragment Analysis -> Intensity Histogram plot."*
4. **Number of b-type ions is less than three times the count of y-type ions (and visa-versa):** *"b-type ions significantly outnumber y-type ions. This may be due to: 1) Incorrect C-terminal sequence, modification state, or C-terminus. 2) An uneven charge distribution. To check the distribution of charges along the protein, use the Fragment Analysis -> Intensity Histogram plot."*

Together with the native fragmentation propensity score used to support protein identification, these flags guide users in assessing the quality of their assignments and identifying common issues. The utility of this approach is demonstrated in a new entrapment-style analysis, now included in **Supplementary Figure 19, Supplementary Note 2**, and discussed in our response to Reviewer 2, Point 1 on pages 7-8 of our responses.

G. Key early top-down membrane protein papers are not cited. Collision energies were minimized to completely avoid internal fragments.

We have now cited two early papers describing top-down mass spectrometry of membrane proteins to support the observation that b-/y-type ions are the most abundant products at low collision energies.

Whitelegge, J. P., Zhang, H., Aguilera, R., Taylor, R. M. & Cramer, W. A. Full Subunit Coverage Liquid Chromatography Electrospray Ionization Mass Spectrometry (LCMS+) of an

Oligomeric Membrane Protein: Cytochrome b6f Complex From Spinach and the Cyanobacterium *Mastigocladus Laminosus*. *Molecular & Cellular Proteomics* **1**, 816-827 (2002). <https://doi.org/https://doi.org/10.1074/mcp.M200045-MCP200>

Ryan, C. M. *et al.* Post-translational Modifications of Integral Membrane Proteins Resolved by Top-down Fourier Transform Mass Spectrometry with Collisionally Activated Dissociation. *Molecular & Cellular Proteomics* **9**, 791-803 (2010). <https://doi.org/https://doi.org/10.1074/mcp.M900516-MCP200>

H. The body text (results/discussion) is somewhat fuzzy and fails to emphasize the points that are dealt with in more detail in methods/supplemental sections.

We acknowledge that the body text does not contain extensive experimental details. We structured our manuscript with the broad readership of Nature Methods in mind, focussing the Results and Discussion sections on the key conceptual innovations and biological findings while providing comprehensive details in the Methods section. This approach was intended to make our work accessible to researchers across different disciplines while ensuring that specialists could still access the full methodological details they require. To maintain such general readability, we have retained the technical details in the Methods section.

Response to Reviewers' Comments:

Reviewers' Comments:

Reviewer #1 (Remarks to the Author):

With this revised manuscript, the authors have made a sincere effort to address my concerns, including by performing additional experiments and analysis. I believe the precision software package will have an important impact in native as well as denaturing TDMS.

Reviewer #2 (Remarks to the Author):

The authors have put forth considerable effort to address my comments, and I believe they have improved both their approach and the manuscript. I see this work as now suitable for publication.

Remarks on figshare data availability:

This appears to be high quality data in line with claims in the manuscript.

Reviewer #3 (Remarks to the Author):

The authors have addressed all concerns in a satisfactory manner, and the paper is suitable for publication. The topic is timely, and the reported strategy is compelling. I anticipate that the paper will generate significant interest from the mass spectrometry community.

Reviewer #4 (Remarks to the Author):

A. The manuscript details an advanced algorithm for assignment of top-down MSMS data that extends the state of the art significantly.

B. The algorithm is sufficiently advanced compared to previous programs that it can be regarded as original and significant.

C. The figures (main and supplementary) demonstrate the validity of the approach and the quality of data produced on current advanced instrumentation.

D. I am happy with use of statistics and treatment of uncertainties.

E. The conclusions are robust and reliable. I see no reason to doubt the validity of the conclusions.

F. Though beyond the scope of this manuscript I hope that future studies will use truly native proteins rather than those with sequences modified by mutagenesis and expressed in cell lines.

G. Appropriate credit was given to previous work in the references.

H. Clarity and context was excellent. The lucidity and appropriateness of the abstract, introduction and conclusions was very good.

Remarks on code availability:

I do not have credentials to review code.

We thank the reviewers for their kind comments.